# LEARNING POST-NONLINEAR CAUSAL RELATIONSHIP WITH FINITE SAMPLES: A MAXIMAL CORRELATION PERSPECTIVE

## ABSTRACT

Bivariate causal discovery aims to determine the causal relationship between two random variables from passive observational data (as intervention is not affordable in many scientific fields), which is considered fundamental and challenging. Designing algorithms based on the post-nonlinear (PNL) model has aroused much attention for its generality. However, the state-of-the-art (SOTA) PNL-based algorithms involve highly non-convex objectives due to the use of neural networks and non-convex losses, thus optimizing such objectives is often time-consuming and unable to produce meaningful solutions with finite samples. In this paper, we propose a novel method that incorporates maximal correlation into the PNL model learning (short as MC-PNL) such that the underlying nonlinearities can be accurately recovered. Owing to the benign structure of our objective function, when modeling the nonlinearities with linear combinations of random Fourier features, the target optimization problem can be solved rather efficiently and rapidly via the block coordinate descent. We also compare the MC-PNL with SOTA methods on the downstream synthetic and real causal discovery tasks to show its superiority in time and accuracy. Our code is available at https://anonymous.4open.science/r/MC-PNL-3C09/ .

## 1 INTRODUCTION & RELATED WORKS

Causal discovery has recently gained significant attention within the machine learning community, which aims to find causal relationships among variables. Many recent attempts at application have emerged in various scientific domains, such as climate science (Ebert-Uphoff & Deng, 2012; Runge et al., 2019), bioinformatics (Choi et al., 2020; Foraita et al., 2020; Shen et al., 2020), etc. The gold standard for causal discovery is to conduct randomized experiments (via interventions), however, interventions are often expensive and impractical. It is highly demanded to discover causal relationships purely from passive observational data. In the past three decades, many pioneer algorithms for directed acyclic graph (DAG) searching have been developed for multivariate causal discovery to reduce the computational complexity and improve the accuracy. For example, there are constraint/independence-based algorithms such as IC, PC, FCI (Pearl, 2009; Spirtes et al., 2000), RFCI (Colombo et al., 2012) (too many to be listed), as well as score-based methods such as GES (Chickering, 2002), NOTEARS (Zheng et al., 2018), etc. However, the algorithms mentioned above can merely return a Markov equivalence class (MEC) that encodes the same set of conditional independencies, with many undetermined edge directions. In this paper, we will focus on a fundamental problem, namely bivariate causal discovery, which aims to determine the causal direction between two random variables $X$ and $Y$. Bivariate causal discovery is one promising routine for appropriate identification of the underlying *causal* DAG (Peters et al., 2017).

Bivariate causal discovery is a challenging task, which cannot be directly solved using the existing methodologies for the multivariate case, because the two candidate DAGs, $X \rightarrow Y$ and $X \leftarrow Y$, are in the same MEC. To make bivariate causal discovery feasible, it is necessary to impose additional assumptions, as summarized in Peters et al. (2017). One vital assumption is on the *a priori* restricted model class, e.g., linear non-Gaussian acyclic model (LiNGAM) (Shimizu et al., 2006), nonlinear additive noise model (ANM) (Mooij et al., 2016), post-nonlinear (PNL) model (Zhang & Hyvärinen, 2009), etc. The other assumption is on the "independence of cause and mechanism" leading to the

algorithms of trace condition (Janzing et al., 2010), IGCI (Janzing et al., 2012), distance correlations (Liu & Chan, 2016), meta-transfer (Bengio et al., 2020), CDCI (Duong & Nguyen, 2022), etc. There are also seminal works focusing on causal discovery in linear/nonlinear dynamic systems, which are out of the scope of this paper, and the corresponding representatives are the Granger causality test (Granger, 1969) and convergent cross mapping (Sugihara et al., 2012).

In this work, we focus on the PNL model, given its generality compared to LiNGAM and ANM. The PNL learning problem is essentially a bi-variate case of the PNL mixture separation (Taleb & Jutten, 1999). The existing works merely show the identifiability results with infinite data samples, while practical issues with finite sample size are seldom discussed. In this paper, we will reveal the difficulties with the current PNL-based algorithms in the finite sample regime, such as insufficient model fitting, slow training progress, and unsatisfactory independent test performance, and correspondingly propose novel and practical solutions.

The main contributions of this work are as follows.

1. We systematically **discuss the pros and cons** of the existing PNL model learning algorithms, in particular the independence-based and maximal correlation-based algorithms, and **propose a new algorithm called MC-PNL** (specifically, the *maximal correlation*-based algorithm with *independence regularization*), which can **achieve a better nonlinear transformation recovery** with finite samples.

2. The devised MC-PNL objective admits a **benign optimization structure** and can be optimized with the block coordinate descent (BCD) algorithm efficiently.

3. We suggest using the randomized dependence coefficient (RDC) instead of the Hilbert-Schmidt independence criterion (HSIC) for the independence test with finite samples, and give **a universal view of a subset of widely used dependence measures** from the perspective of *squared-loss mutual information* estimation.

4. We use MC-PNL in bivariate causal discovery and show that our method outperforms other SOTAs on various benchmark datasets (**20-300$\times$ faster** with competitive accuracy).

## 2 PRELIMINARIES

In this section, we will introduce the HSIC as a dependence measure for regression, the current independence test-based causal discovery methods for PNL model, and other relevant learning methods based on the Hirschfeld-Gebelein-Rényi (HGR) correlation. Our proposed MC-PNL method exploits all these ingredients.

### 2.1 HSIC-BASED REGRESSION

Regression by dependence minimization (Mooij et al., 2009) has recently attracted much attention and shown its power for robust learning (Greenfeld & Shalit, 2020). Consider the additive noise model (ANM), $Y = f(X) + \epsilon, \epsilon \perp\!\!\!\perp X$, where the additive noise $\epsilon$ is assumed to be independent with the input variable $X$. The selected regression model $f_{\boldsymbol{\theta}}$ is to be learned via **minimizing the dependence** between the input variable $X$ and the residual $Y - f_{\boldsymbol{\theta}}(X)$. A widely used dependence measure is the Hilbert-Schmidt independence criterion (HSIC) (Gretton et al., 2005; 2007).

**Definition 1** (HSIC). *Let $X, Z \sim P_{XZ}$ be jointly distributed random variables, and $\mathcal{F}, \mathcal{G}$ be reproduced kernel Hilbert spaces with kernel functions $k(\cdot, \cdot)$ and $l(\cdot, \cdot)$, the HSIC is expressed as,*

$$\begin{aligned}
\mathrm{HSIC}(X, Z; \mathcal{F}, \mathcal{G}) = {}& \mathbb{E}_{XZ}\mathbb{E}_{X'Z'}k\left(x, x'\right)l\left(z, z'\right) + \mathbb{E}_X\mathbb{E}_{X'}k\left(x, x'\right)\mathbb{E}_Z\mathbb{E}_{Z'}l\left(z, z'\right) \\
& - 2\mathbb{E}_{X'Z'}\left[\mathbb{E}_X k\left(x, x'\right)\mathbb{E}_Z l\left(z, z'\right)\right],
\end{aligned} \tag{1}$$

*where $x'$ and $z'$ denote independent copies of $x$ and $z$, respectively.*

**Remark 2.1.** *We can conclude that: (a) $X \perp\!\!\!\perp Z \Rightarrow \mathrm{HSIC}(X, Z) = 0$; (b) with a proper universal kernel (e.g., Gaussian kernel), $X \perp\!\!\!\perp Z \Leftarrow \mathrm{HSIC}(X, Z) = 0$ (Gretton et al., 2005).*

When the joint $P_{XZ}$ is unknown, given a dataset with $n$ samples ($\boldsymbol{x} = [x_1, x_2, \ldots, x_n]^T \in \mathbb{R}^n, \boldsymbol{z} = [z_1, z_2, \ldots, z_n]^T \in \mathbb{R}^n$), a biased HSIC estimate can be constructed as,

$$\widehat{\mathrm{HSIC}}\left(\boldsymbol{x}, \boldsymbol{z}; \mathcal{F}, \mathcal{G}\right) = \frac{1}{n^2}\operatorname{tr}(KHLH) = \frac{1}{n^2}\langle HKH, HLH\rangle, \tag{2}$$

where $K_{i,j} = k(x_i, x_j)$, $L_{i,j} = l(z_i, z_j)$, and $H = I - \frac{1}{n}\mathbf{1}\mathbf{1}^T \in \mathbb{R}^{n \times n}$ is a centering matrix. The Gaussian kernel $k(x_i, x_j) = \exp\left(-(x_i - x_j)^2 \sigma^{-2}\right)$ is commonly used, and the same for $l$. This empirical HSIC can be interpreted as the inner-product of two centralized kernel matrices that summarize the sample similarities.

Mooij et al. (2009) first proposed to use the empirical HSIC (2) for ANM learning. Concretely, the regression model is a linear combination of the basis functions, $\phi_i(\cdot)$, $i = 1, 2, ..., k$, namely $f_{\boldsymbol{\theta}}(x) = \sum_{i=1}^{k} \theta_i \phi_i(x)$; and the parameters, $\boldsymbol{\theta} = [\theta_1, ..., \theta_k]^T$, are learned from:

$$\hat{\boldsymbol{\theta}} \in \underset{\boldsymbol{\theta} \in \mathbb{R}^k}{\arg\min} \left( \widehat{\mathrm{HSIC}}(\boldsymbol{x}, \boldsymbol{y} - f_{\boldsymbol{\theta}}(\boldsymbol{x})) + \frac{\lambda}{2}\|\boldsymbol{\theta}\|_2^2 \right), \tag{3}$$

where $f_{\boldsymbol{\theta}}$ is applied element-wisely to the data points, and $\lambda > 0$ is a penalty parameter (we will keep using $\lambda$ as a penalty parameter under different contexts). One key advantage of this formulation is that it does not require any assumption on the noise distribution. Greenfeld & Shalit (2020) further implemented $f_{\boldsymbol{\theta}}$ using neural networks, and showed the learnability with the HSIC loss theoretically.

## 2.2 CAUSAL DISCOVERY WITH PNL MODEL

Compared to ANM, the PNL model is preferred due to its richer representation power. The bivariate PNL model is expressed as, $Y = f_2(f_1(X) + \epsilon)$, where $f_1$ denotes the nonlinear effect of the cause, $\epsilon$ is the independent noise, and $f_2$ denotes the invertible post-nonlinear distortion from the sensor or measurement side. The goal is to find the causal direction $X \to Y$ from a set of passive observations on $X$ and $Y$, assuming no unobserved confounders. Note that from the data generating process, $\epsilon$ is independent with $X$ but not $Y$. Taking this asymmetry as a prior, one can determine the causal direction by first learning the underlying transformations, $f_2^{-1}$ and $f_1$, and then testing the independence between the residual $r_{(\to)} = f_2^{-1}(Y) - f_1(X)$ and the input $X$.

The PNL-MLP algorithm proposed by Zhang & Hyvärinen (2009) tests between two hypotheses ($X \to Y$ and $X \leftarrow Y$) as follows. Under the hypothesis $X \to Y$, one can parameterize $f_1$ and $f_2^{-1}$ by two multi-layer perceptrons (MLPs) $f_{(\to)}$ and $g_{(\to)}$, and learn them via minimizing the mutual information (MI):

$$\hat{f}_{(\to)}, \hat{g}_{(\to)} \in \underset{f_{(\to)}, g_{(\to)}}{\arg\min} \mathrm{MI}\left(\boldsymbol{r}_{(\to)}; \boldsymbol{x}\right), \tag{4}$$

where $\boldsymbol{r}_{(\to)} := g_{(\to)}(\boldsymbol{y}) - f_{(\to)}(\boldsymbol{x})$, and $g_{(\to)}, f_{(\to)}$ are applied element-wisely. The resulting estimated residual is $\hat{\boldsymbol{r}}_{(\to)} = \hat{g}_{(\to)}(\boldsymbol{y}) - \hat{f}_{(\to)}(\boldsymbol{x})$. Similarly, under the hypothesis $X \leftarrow Y$, one can obtain an estimate of $\hat{\boldsymbol{r}}_{(\leftarrow)} = \hat{g}_{(\leftarrow)}(\boldsymbol{x}) - \hat{f}_{(\leftarrow)}(\boldsymbol{y})$ via minimizing $\mathrm{MI}(\boldsymbol{r}_{(\leftarrow)}; \boldsymbol{y})$. The causal direction is determined by comparing $\widehat{\mathrm{HSIC}}\left(\hat{\boldsymbol{r}}_{(\to)}, \boldsymbol{x}\right)$ and $\widehat{\mathrm{HSIC}}\left(\hat{\boldsymbol{r}}_{(\leftarrow)}, \boldsymbol{y}\right)$. If $\widehat{\mathrm{HSIC}}\left(\hat{\boldsymbol{r}}_{(\to)}, \boldsymbol{x}\right) < \widehat{\mathrm{HSIC}}\left(\hat{\boldsymbol{r}}_{(\leftarrow)}, \boldsymbol{y}\right)$, the hypothesis $X \to Y$ is endorsed; otherwise, the hypothesis $X \leftarrow Y$ is endorsed.

However, the MI between random variables is often difficult to calculate (see suppl. A), and tuning the MLPs requires many tricks as mentioned in Zhang & Hyvärinen (2009). The PNL-MLP algorithm may easily fall into some undesired local optima. To eliminate those, Uemura & Shimizu (2020) proposed the AbPNL algorithm that directly uses HSIC instead of MI as the objective, and imposes the invertibility constraint of $f_2$ via an auto-encoder,

$$\min_{f, g, g'} \widehat{\mathrm{HSIC}}\left(g(\boldsymbol{y}) - f(\boldsymbol{x}), \boldsymbol{x}\right) + \lambda\|\boldsymbol{y} - g'(g(\boldsymbol{y}))\|_2^2, \tag{5}$$

where $g, g'$ are the encoder and decoder MLPs. The subscript $_{(\to)}$ is omitted for conciseness here. The learning architectures of the above-mentioned two methods are summarized in Figure 1. Nevertheless, inherent issues still exist, concerning the cost function and the neural network training procedure when dealing with finite samples, see Section 3. Recently, Keropyan et al. (2023) proposed a rank-based method (rank-PNL), and showed its efficacy in PNL learning. We also include it as a baseline.

## 2.3 PNL LEARNING THROUGH MAXIMAL CORRELATION

A more generic and effective routine to learn the nonlinear transformations $f$ and $g$ is through the HGR maximal correlation (Rényi, 1959).

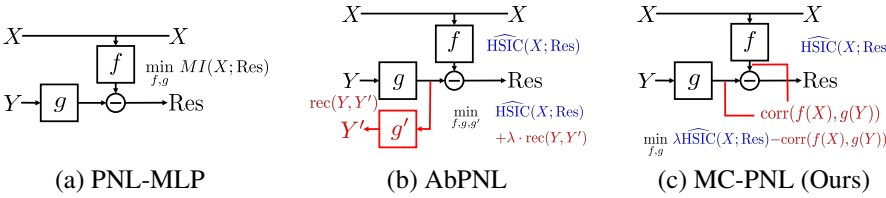

Figure 1: Comparisons of several PNL model learning architectures.

**Definition 2** (HGR maximal correlation). *Let $X, Y$ be jointly distributed random variables. Then,*

$$\rho^* = \mathrm{HGR}(X; Y) := \sup_{\substack{f:\mathcal{X}\to\mathbb{R}, g:\mathcal{Y}\to\mathbb{R} \\ \mathbb{E}[f(X)]=\mathbb{E}[g(Y)]=0 \\ \mathbb{E}[f^2(X)]=\mathbb{E}[g^2(Y)]=1}} \mathbb{E}[f(X)g(Y)], \tag{6}$$

*is the HGR maximal correlation between $X$ and $Y$, where $f, g$ are the associated transformations.*

**Remark 2.2.** *The HGR maximal correlation $\rho^*$ is attractive as a measure of dependency due to some useful properties: (1) Bounded $\rho^*$ : $0 \le \rho^* \le 1$; (2) $X$ and $Y$ are independent if and only if $\rho^* = 0$.*

The optimal unit-variance feature transformations, $f^*$ and $g^*$, can be found by iteratively updating $f$ and $g$ in (6). However, for causal discovery applications, one fatal issue of using the HGR is that the unit-variance $f^*$ and $g^*$ cannot reflect the true magnitudes of the underlying functions $f$ and $g$. As a consequence, the resulting residual can be erroneous for the independence tests in the next stage. We found two feasible remedies in the literature, namely the alternating conditional expectation (ACE) algorithm (Breiman & Friedman, 1985) and a soft version of (6) (Soft-HGR)(Wang et al., 2019).

The ACE algorithm solves the regression problem (7) by computing the conditional mean alternatively,

$$\begin{aligned} \min_{f,g} \quad & \mathbb{E}(f(X) - g(Y))^2, \\ \text{s.t.} \quad & \mathbb{E}[f(X)] = \mathbb{E}[g(Y)] = 0, \mathbb{E}[g^2(Y)] = 1, \end{aligned} \tag{7}$$

which imposes the unit-variance constraint only on $g$. Problem (7) is equivalent to (6), and the optimal transformation pair $(f^*_{\mathrm{ACE}}, g^*_{\mathrm{ACE}})$ equals $(\rho^* f^*, g^*)$ (Breiman & Friedman, 1985).

The other formulation, Soft-HGR, relaxes the unit-variance constraints as follows,

$$\begin{aligned} \max_{f,g} \quad & \mathbb{E}\left[f(X)g(Y)\right] - \frac{1}{2}\mathrm{var}(f(X))\,\mathrm{var}(g(Y)), \\ \text{s.t.} \quad & \mathbb{E}[f(X)] = \mathbb{E}[g(Y)] = 0, \end{aligned} \tag{8}$$

which allows linearly transformed solutions $(af^*, a^{-1}g^*), \forall a \in \mathbb{R}\backslash\{0\}$. This scale ambiguity results in enormous solutions, out of which the desired one for causal discovery must enforce the estimated residual to be independent of the input. We notice that the cutting-edge method in self-supervised learning, i.e., Variance-Invariance-Covariance Regularization (Bardes et al., 2022), actually shares the same rationale with the HGR maximal correlation, see suppl. B for more details.

## 3 PROS & CONS OF EXISTING ALGORITHMS

In this section, we summarize the pros and cons of the existing PNL learning algorithms, including among others PNL-MLP, AbPNL, and ACE, which motivate our proposed MC-PNL to be introduced in Section 4. We also discuss the choice of dependence measures in the finite sample regime, as it is an important module for causal discovery.

### 3.1 PROS & CONS OF EXISTING PNL LEARNING ALGORITHMS

INDEPENDENCE-BASED METHOD   The learning objectives of PNL-MLP and AbPNL match the independent noise assumption, yet they suffer from the following over-fitting and optimization issues.

**Over-fitting issue.** The general idea of PNL learning, according to Section 2.2, is to encourage statistical independence between the input and the residual. Both PNL-MLP and AbPNL use neural networks to parameterize $f$ and $g$. However, it is doubtful that meaningful representations can be learned with finite samples. To reveal this, let us review the dependence minimization problem below,

$$\min_{f,g}\{\widehat{\mathrm{HSIC}}(\boldsymbol{x},\boldsymbol{r}) := \frac{1}{n^2}\operatorname{tr}(K_{\boldsymbol{xx}}HL_{\boldsymbol{rr}}H)\}, \tag{9}$$

where $\boldsymbol{r} = f(\boldsymbol{x}) - g(\boldsymbol{y})$ is the residual term. We argue that it is utmost difficult to learn meaningful representations of $f$ and $g$ via minimizing solely the HSIC score, due to the enormous degrees of freedom for $f$ and $g$ to fit arbitrary random noise profiles. We adopted *wide over-parameterized* and *narrow deep* neural networks for $f$ and $g$ in simulations, and they both can achieve zero training loss but unfortunately produce meaningless estimates, see suppl. C. This is unsurprising though, as one can force $\boldsymbol{r}$ to match samples from arbitrary independent random noise (Zhang et al., 2021). To resolve with this issue, we propose to cooperate dependence minimization with maximal correlation, which helps to obtain desired solutions, see Figure 1(c) for illustration and Section 4 for details.

**Optimization issue.** Efficient optimization of neural networks is a long-standing problem, and yet there is not any study on the optimization landscape of the HSIC loss with neural networks. Typically, first-order methods such as stochastic gradient descent were used in the existing causal discovery methods, and the goodness of initialization is crucial to the causal discovery accuracy, see suppl. C. *In this paper, we propose to parameterize both $f$ and $g$ as a linear combination of random Fourier features and adopt a linear kernel for HSIC, such that the resulting non-convex optimization problem has a benign landscape with symmetry* (see Chapter 7 in Wright & Ma (2022)).

MAXIMAL CORRELATION-BASED METHODS    For ACE and Soft-HGR, their objective functions are often easier to optimize with proper parameterization, and can produce meaningful solutions. However, the independence property is not ensured, see the following inconsistency issue for the ACE model in (7).

**Lemma 1** (Inconsistency of ACE). *Suppose the data were generated from the PNL model $Y = f_2(f_1(X) + \epsilon)$, where $\epsilon \perp\!\!\!\perp X$. And let $\bar{g}(Y) = \frac{f_2^{-1}(Y) - \mathbb{E}[f_2^{-1}(Y)]}{\sqrt{\mathbb{E}[f_2^{-1}(Y) - \mathbb{E}[f_2^{-1}(Y)]]^2}}$ and $\bar{f}(X) = \frac{f_1(X) - \mathbb{E}[f_1(X)]}{\sqrt{\mathbb{E}[f_2^{-1}(Y) - \mathbb{E}[f_2^{-1}(Y)]]^2}}$ be the underlying ground truth of (7). Then $(\bar{f}, \bar{g})$ is not a local minimum of the regression problem (7).*

It is not hard to show, starting from $(\bar{f}, \bar{g})$, the function $g$ can always be further optimized to improve the objective value. A similar result also holds for Soft-HGR. The objective of ACE or Soft-HGR failed to capture the asymmetricity induced by the independent causal mechanism. Thus, the resulting solution will always be deviated/distorted from the underlying $(\bar{f}, \bar{g})$. It is necessary to involve the independence constraint newly introduced in Section 4.

## 3.2 DISCUSSIONS ON INDEPENDENCE TEST

As the independence test is crucial to the causal discovery accuracy, we shall cautiously choose a dependence measure. Although the HSIC is widely used, there are several drawbacks (e.g., the choice of kernel and corresponding hyper-parameters are user-defined, the HSIC value depends on the scale of the random variables). It is shown experimentally that the HSIC score may not be a proper choice, and we **favor randomized dependence coefficient (RDC)** (Lopez-Paz et al., 2013), particularly **for finite samples**.

We generate various synthetic datasets following the PNL models (see suppl. D), and know in advance that $\epsilon \perp\!\!\!\perp X$ and $\epsilon \not\!\perp\!\!\!\perp Y$. Thus, we are able to compare various dependence measures, by checking whether $\mathrm{Dep}(\boldsymbol{x}, \boldsymbol{\epsilon}) < \mathrm{Dep}(\boldsymbol{y}, \boldsymbol{\epsilon})$ holds for all datasets. Herein, the compared dependence measures are HSIC (Gretton et al., 2005), its normalized variant (NOCCO) (Fukumizu et al., 2007), and RDC (Lopez-Paz et al., 2013). Besides, we also studied the impact of different choices of linear, Gaussian radial basis function (RBF), and rational quadratic (RQ) kernels. We stress especially that RDC is a computationally tractable estimator inspired by the HGR maximal correlation, and it outperforms other dependence measures especially when the sample size is small, see Table 1.

Table 1: The independence test accuracy (%) with known injected noise.

| # of samples | $n = 500$ | | | | $n = 1000$ | | | | $n = 2000$ | | | | $n = 5000$ | | | |
|---|---|---|---|---|---|---|---|---|---|---|---|---|---|---|---|---|
| noise level, $\sigma_\epsilon$ | 0.01 | 0.1 | 1 | 10 | 0.01 | 0.1 | 1 | 10 | 0.01 | 0.1 | 1 | 10 | 0.01 | 0.1 | 1 | 10 |
| HSIC-linear | 64 | 81 | 95 | 100 | 69 | 79 | 95 | 100 | 67 | 88 | 96 | 100 | 73 | 85 | 97 | 100 |
| HSIC-RBF | 64 | 88 | 99 | 100 | 66 | 85 | 100 | 100 | 75 | 95 | 100 | 100 | 77 | 99 | 100 | 100 |
| HSIC-RQ | 77 | **93** | **100** | **100** | 81 | 92 | **100** | **100** | 86 | **100** | **100** | **100** | 86 | **100** | **100** | **100** |
| NOCCO-RBF | 65 | 82 | 91 | 97 | 73 | 88 | 94 | 99 | 73 | 89 | 98 | 100 | 76 | 92 | 99 | 100 |
| NOCCO-RQ | 66 | 84 | 91 | 91 | 67 | 84 | 94 | 98 | 68 | 85 | 95 | 99 | 75 | 91 | 96 | 100 |
| **RDC** | **90** | **93** | **100** | **100** | **84** | **94** | **100** | **100** | **91** | 98 | **100** | **100** | **87** | **100** | **100** | **100** |

We also provide **a universal view of the aforementioned dependence measures** through *squared-loss mutual information* (SMI) (Suzuki et al., 2009), see our derivations in the suppl. E.

## 4 PROPOSED METHOD

In this section, we propose a new maximal correlation-based post-nonlinear model learning algorithm, called MC-PNL, to accurately estimate the nonlinear functions and compute the corresponding residuals. Thereafter, independence tests will be conducted to determine the causal direction.

### 4.1 MAXIMAL CORRELATION-BASED PNL LEARNING WITH INDEPENDENCE CONSTRAINT

As we have seen in Section 3, minimizing HSIC (9) requires no assumption on the noise distribution and encourages independent residuals, but it can easily get stuck at meaningless local minima. Maximal correlation-based methods can learn meaningful transformations as the name suggested, but do not necessarily produce independent residuals. To combine their strengths, we propose the following MC-PNL formulation that imposes the independence constraint to (8),

$$\min_{f,g} \quad -\mathbb{E}\left[f(X)g(Y)\right] + \frac{1}{2}\operatorname{var}(f(X))\operatorname{var}(g(Y)),$$
$$\text{s.t.} \quad \mathbb{E}[f(X)] = \mathbb{E}[g(Y)] = 0, \quad \operatorname{Dep}(X, f(X) - g(Y)) = 0, \tag{10}$$

where $\operatorname{Dep}(\cdot, \cdot) \geq 0$ is a dependence measure (e.g., HSIC with different kernel functions). The hard independence constraint can ensure the recovery of ground truth up to a linear transformation.

**Lemma 2.** *Assuming invertible $\bar{f}$ and $\bar{g}$, we have*

$$\operatorname{HSIC}(f(X) - g(Y), X) = 0 \quad \Rightarrow \quad f(X) = a\bar{f}(X) + b_1 \text{ and } g(Y) = a\bar{g}(Y) + b_2, \tag{11}$$

*where $a \neq 0, b_0, b_1$ are some constant numbers.*

**Proposition 1.** *Assuming invertible $\bar{f}$ and $\bar{g}$, (10) has only two solutions aligning with the underlying ground truth up to a sign ambiguity, i.e., $(a^*\bar{f}, a^*\bar{g})$, where $a^* = \pm\sqrt{\frac{\mathbb{E}[\bar{f}(X)\bar{g}(Y)]}{\mathbb{E}(\bar{f}^2(X))\mathbb{E}(\bar{g}^2(Y))}}$.*

The proof can be found in the suppl. F. While the independence constraint is non-convex and difficult to deal with, we adopt the following penalized form instead,

$$\min_{f,g} \quad -\mathbb{E}\left[f(X)g(Y)\right] + \frac{1}{2}\operatorname{var}(f(X))\operatorname{var}(g(Y)) + \lambda\operatorname{Dep}(X, f(X) - g(Y)),$$
$$\text{s.t.} \quad \mathbb{E}[f(X)] = \mathbb{E}[g(Y)] = 0, \tag{12}$$

where $\lambda > 0$ is a newly introduced hyper-parameter. We aim to learn meaningful feature transformations with the Soft-HGR term, and resolve the scale ambiguity via the dependence penalty.

**Parameterization with Random Fourier Features**

For ease of optimization, we parameterize the transformation functions as the linear combination of the random Fourier features, namely $f(x; \boldsymbol{\alpha}) := \boldsymbol{\alpha}^T\boldsymbol{\phi}(x)$ and $g(y; \boldsymbol{\beta}) := \boldsymbol{\beta}^T\boldsymbol{\psi}(y)$ , where the random Fourier features $\boldsymbol{\phi}(x) \in \mathbb{R}^{k_1}, \boldsymbol{\psi}(y) \in \mathbb{R}^{k_2}$ are nonlinear projections as described in Lopez-Paz et al. (2013), see suppl. G. For a given dataset $\{(x_i, y_i)\}_{i=1}^n$, the corresponding feature matrices are denoted as $\Phi := [\boldsymbol{\phi}(x_1), \boldsymbol{\phi}(x_2), \ldots, \boldsymbol{\phi}(x_n)] \in \mathbb{R}^{k_1 \times n}$ and $\Psi := [\boldsymbol{\psi}(y_1), \boldsymbol{\psi}(y_2), \ldots, \boldsymbol{\psi}(y_n)] \in \mathbb{R}^{k_2 \times n}$. We further denote the residual vector as $\boldsymbol{r} := \Phi^T\boldsymbol{\alpha} - \Psi^T\boldsymbol{\beta}$.

Consequently, (12) can be written as the following non-convex optimization problem,

$$\min_{\boldsymbol{\alpha},\boldsymbol{\beta}} \quad J(\boldsymbol{\alpha},\boldsymbol{\beta}) := -\frac{1}{n}\boldsymbol{\alpha}^T\Phi\Psi^T\boldsymbol{\beta} + \frac{1}{2n^2}\boldsymbol{\alpha}^T\Phi\Phi^T\boldsymbol{\alpha}\boldsymbol{\beta}^T\Psi\Psi^T\boldsymbol{\beta} + \lambda\mathrm{Dep}(\boldsymbol{x},\boldsymbol{r}),$$

$$\text{s.t.} \quad \boldsymbol{\alpha}^T\Phi\mathbf{1} = \boldsymbol{\beta}^T\Psi\mathbf{1} = 0, \tag{13}$$

where the dependence measure, $\mathrm{Dep}(\boldsymbol{x},\boldsymbol{r})$, can be specially set to the HSIC with linear kernel, namely,

$$\widehat{\mathrm{HSIC}}^{lin}(\boldsymbol{x},\boldsymbol{r}) = \frac{1}{n^2}\mathrm{tr}(K_{\boldsymbol{xx}}HL_{\boldsymbol{rr}}^{lin}H) = \frac{1}{n^2}\mathrm{tr}(K_{\boldsymbol{xx}}H\boldsymbol{r}\boldsymbol{r}^TH)$$

$$= \frac{1}{n^2}(\boldsymbol{\alpha}^T\Phi HK_{\boldsymbol{xx}}H\Phi^T\boldsymbol{\alpha} + \boldsymbol{\beta}^T\Psi HK_{\boldsymbol{xx}}H\Psi^T\boldsymbol{\beta} - 2\boldsymbol{\alpha}^T\Phi HK_{\boldsymbol{xx}}H\Psi^T\boldsymbol{\beta}). \tag{14}$$

**Remark:** We adopt the HSIC with linear kernel $L_{\boldsymbol{rr}}^{lin} := \boldsymbol{r}\boldsymbol{r}^T$ mainly for a favorable optimization structure, as the resulting HSIC score admits a quadratic form with respect to (w.r.t.) both $\boldsymbol{\alpha}$ and $\boldsymbol{\beta}$. Note that the penalty HSIC term is always non-negative, but the Soft-HGR objective can be negative.

The above problem can be solved via the famous BCD algorithm that updates $\boldsymbol{\alpha}$ and $\boldsymbol{\beta}$ iteratively, see Algorithm 1. In each update (line 3 or 4), the sub-problem belongs to linearly constrained quadratic programs. When sub-problems are strictly convex, unique minimum in closed-form can be obtained (see suppl. H), ensuring convergence to a critical point (Grippo & Sciandrone, 2000).

| **Algorithm 1** `BCD` for problem (13) | **Algorithm 2** `MC-PNL` for causal direction inference. |
|---|---|
| 1: Initialize $\boldsymbol{\alpha}^{(0)}$ and $\boldsymbol{\beta}^{(0)}$ 
 2: **for** $t \leftarrow 1$ **to** $T$ **do** 
 3:   $\boldsymbol{\alpha}^{(t)} \leftarrow \arg\min_{\boldsymbol{\alpha}} J(\boldsymbol{\alpha},\boldsymbol{\beta}^{(t-1)})$, 
    subject to $\boldsymbol{\alpha}^T\Phi\mathbf{1} = 0$. 
 4:   $\boldsymbol{\beta}^{(t)} \leftarrow \arg\min_{\boldsymbol{\beta}} J(\boldsymbol{\alpha}^{(t)},\boldsymbol{\beta})$, 
    subject to $\boldsymbol{\beta}^T\Psi\mathbf{1} = 0$. 
 5:   **if** stopping criteria are met **then** 
 6:    **return** $\boldsymbol{\alpha}^{(t)},\boldsymbol{\beta}^{(t)}$ 
 7:   **end if** 
 8: **end for** | **Input:** Standardized data $\boldsymbol{x},\boldsymbol{y} \in \mathbb{R}^n$, decision criterion $\delta$. 
 **Output:** The causal direction with $C_{X\to Y}$. 
 1: Fit PNL models via Algorithm 1 and estimate residuals under hypotheses, $H_0 : X \to Y$ and $H_1 : X \leftarrow Y$. 
  - Under $H_0$, $\hat{\boldsymbol{r}}_{(\to)} = \hat{g}_{(\to)}(\boldsymbol{y}) - \hat{f}_{(\to)}(\boldsymbol{x})$. 
  - Under $H_1$, $\hat{\boldsymbol{r}}_{(\leftarrow)} = \hat{g}_{(\leftarrow)}(\boldsymbol{x}) - \hat{f}_{(\leftarrow)}(\boldsymbol{y})$. 
 2: Compute $C_{X\to Y} := \widehat{\mathrm{Dep}}\left(\hat{\boldsymbol{r}}_{(\leftarrow)},\boldsymbol{y}\right) - \widehat{\mathrm{Dep}}\left(\hat{\boldsymbol{r}}_{(\to)},\boldsymbol{x}\right)$. 
 3: Output the causal score $C_{X\to Y}$ and the direction is $X \to Y$ if $C_{X\to Y} > \delta$, $X \leftarrow Y$ if $C_{X\to Y} < \delta$, and no decision if $|C_{X\to Y}| \le \delta$. |

**Residual-Plot Aided Fine-Tuning:** Algorithm 1 may produce solutions with distortions, see Figure 2(a,b), as the independence constraint is softened and a simple linear kernel is used (Gretton et al., 2005). To cope with that, one can enlarge the penalty of the dependence term via $\lambda$, and use HSIC with universal kernels (e.g., RBF kernel) or other dependence measures. Alternatively, as the injected noises are assumed to be independently and identically distributed, implying a horizontal-band residual plot, we design a **banded residual loss** to fine-tune the models as follows. The data samples are separated into $b$ bins $\{\boldsymbol{x}^{(i)},\boldsymbol{y}^{(i)}\}_{i=1}^b$ according to the ordering of $X$, and we expect the residuals in those bins $\mathrm{Res}_i = f(\boldsymbol{x}^{(i)}) - g(\boldsymbol{y}^{(i)})$ to have the same distribution. To this end, we adopt the empirical maximum mean discrepancy (MMD) (Gretton et al., 2012) as a measure of distributional discrepancy. The **banded residual loss** is defined as $\mathrm{band}^{(\mathrm{MMD})} := \sum_{i=1}^b \widehat{\mathrm{MMD}}(\mathrm{Res}_i, \mathrm{Res}_{all})$, where $\mathrm{Res}_{all} = f(\boldsymbol{x}) - g(\boldsymbol{y})$. Then we append this $\mu$-penalized banded loss to (13) as,

$$\min_{\boldsymbol{\alpha},\boldsymbol{\beta}} \quad J(\boldsymbol{\alpha},\boldsymbol{\beta}) + \mu \cdot \mathrm{band}^{(\mathrm{MMD})},$$

$$\text{s.t.} \quad \boldsymbol{\alpha}^T\Phi\mathbf{1} = \boldsymbol{\beta}^T\Psi\mathbf{1} = 0. \tag{15}$$

The fine-tuning problem can be solved by gradient-based algorithms. In general, our proposed method relies on the maximal correlation problem to generate near-optimal solutions while consistently emphasizing the importance of the independence constraint through a readily optimized structure.

### 4.2 Causal Direction Inference via Independence Test

Following the framework proposed by Zhang & Hyvärinen (2009), we infer the causal direction according to Algorithm 2. We first fit nonlinear models $f_{(\to)}, g_{(\to)}$ under hypothesis $X \to Y$, and $f_{(\leftarrow)}, g_{(\leftarrow)}$ under hypothesis $X \leftarrow Y$, using Algorithm 1. Thereafter, we conduct the following

independence tests. If $\widehat{\text{Dep}}\left(\hat{\boldsymbol{r}}_{(\rightarrow)}, \boldsymbol{x}\right) < \widehat{\text{Dep}}\left(\hat{\boldsymbol{r}}_{(\leftarrow)}, \boldsymbol{y}\right)$, the hypothesis $X \rightarrow Y$ is supported; otherwise, the hypothesis $X \leftarrow Y$ is supported. Towards trustworthy decisions, bootstrap (Efron, 1992) can also be used for the uncertainty quantification, see examples in suppl. I.2.

## 5 EXPERIMENTS

In the following, we show the performance of MC-PNL for PNL model learning and bivariate causal discovery applications.

### 5.1 NONLINEAR FUNCTION FITTING

For a better demonstration, we generated two synthetic datasets from the PNL model, $Y = f_2(f_1(X) + \epsilon)$, and each contains 1000 samples. The data generation mechanisms are as follows,

- Syn-1: $f_1(X) = X^{-1} + 10X, f_2(Z) = Z^3, X \sim U(0.1, 1.1), \epsilon \sim U(0, 5)$,
- Syn-2: $f_1(X) = \sin(7X), f_2(Z) = \exp(Z), X \sim U(0, 1), \epsilon \sim N(0, 0.3^2)$.

We apply Algorithm 1 to both datasets and show the learned nonlinear transformations as well as the corresponding residual plots in Figure 2. The underlying nonlinear functions are learned under the true hypothesis but with certain distortions. We also show that, after fine-tuning with HSIC-RBF loss or additionally with our proposed banded residual loss (see suppl. I.1), such distortions can be fixed up. The corresponding residual plot is more of a band shape, and is clearly better than that reported in Uemura & Shimizu (2020).

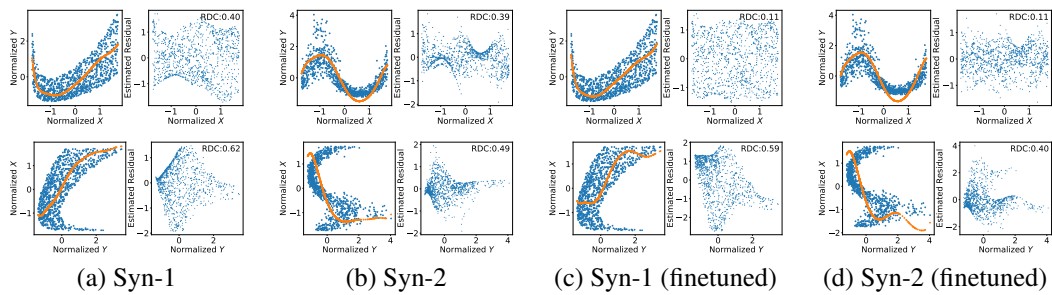

| (a) Syn-1 | (b) Syn-2 | (c) Syn-1 (finetuned) | (d) Syn-2 (finetuned) |

Figure 2: The sub-figures (a) and (b) show the nonlinear function fitting of the two datasets. In each sub-figure, the top row shows the learned $f_{(\rightarrow)}(x)$ (red line) and the residual plot under the correct hypothesis $X \rightarrow Y$, which has a lower RDC value (see top right corner); the bottom row is for the opposite direction, $X \leftarrow Y$. Sub-figures (c) and (d) are the results fine-tuned with HSIC-RBF loss.

**Convergence Results.** We demonstrate the convergence profile of our Algorithm 1 with Syn-2, see Figure 3. Convergence results for Syn-1 can be found in the suppl. I.3. The upper row shows the snapshots of the learned representations, where we do not impose independence regularization ($\lambda = 0$); and Algorithm 1, starting from different random initializations, quickly converges to the local minimizers with the same objective value. The lower row is drawn with independence regularization $\lambda = 5$, where the solutions are identified up to a sign ambiguity.

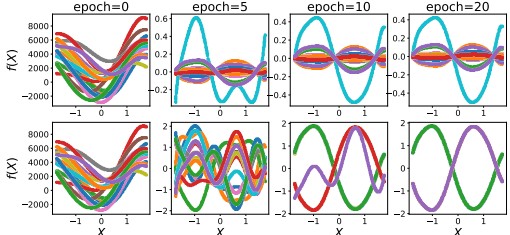

Figure 3: Convergence profile of Algorithm 1 on Syn-2. We plot snapshots of the feature transformations $f$ at the training epochs $0, 5, 10, 20$, using 15 random initializations (indicated by colors). **Upper ($\lambda = 0$):** most initializations converge to local minimizers (symmetry: $(\boldsymbol{\alpha}, \boldsymbol{\beta}) \mapsto (a\boldsymbol{\alpha}, a^{-1}\boldsymbol{\beta})$). **Lower ($\lambda = 5$):** most initializations converge to two local minimizers (symmetry: $(\boldsymbol{\alpha}, \boldsymbol{\beta}) \mapsto -(\boldsymbol{\alpha}, \boldsymbol{\beta})$).

Table 2: Comparison of bivariate causal discovery ROC-AUC on both synthetic and real datasets

| Dataset | ANM[1] (HSIC) | CDS (HSIC) | IGCI | RECI | CDCI | OT-PNL | AbPNL[1] (HSIC) | rank-PNL[1] (HSIC) | ACE[1] (RDC) | MC-PNL[1] (RDC) | MC-PNL[1] (HSIC) |
|---|---|---|---|---|---|---|---|---|---|---|---|
| PNL-A-mixG | 0.256 | 0.207 | 0.932 | 0.537 | 0.410 | 0.431 | 0.645 | 0.635 | 0.580 | 0.708 | 0.702 |
| PNL-B-mixG | 0.150 | 0.160 | 0.908 | 0.462 | 0.304 | 0.309 | 0.672 | 0.388 | 0.536 | 0.771 | 0.738 |
| PNL-A-unif | 0.203 | 0.390 | 0.681 | 0.879 | 0.544 | 0.711 | 0.517 | 0.953 | 0.514 | 0.617 | 0.399 |
| PNL-B-unif | 0.094 | 0.311 | 0.866 | 0.929 | 0.535 | 0.536 | 0.599 | 0.979 | 0.418 | 0.608 | 0.329 |
| D4-S1 | 0.604 | 0.582 | 0.380 | 0.550 | 0.651 | 0.474 | 0.408 | 0.717 | 0.592 | 0.646 | 0.625 |
| D4-S2A | 0.616 | 0.580 | 0.447 | 0.592 | 0.673 | 0.472 | 0.519 | 0.361 | 0.558 | 0.626 | 0.635 |
| D4-S2B | 0.521 | 0.529 | 0.450 | 0.491 | 0.614 | 0.517 | 0.501 | 0.458 | 0.495 | 0.519 | 0.482 |
| D4-S2C | 0.556 | 0.564 | 0.441 | 0.521 | 0.590 | 0.490 | 0.445 | 0.493 | 0.538 | 0.576 | 0.577 |
| Avg. AUC | 0.375 | 0.415 | **0.638** | 0.620 | 0.540 | 0.493 | 0.538 | 0.623 | 0.529 | 0.634 | 0.561 |
| Avg. time[2] (s) | 20.11 | 7.67 | 0.50 | 0.27 | 0.26 | $\sim 7220$ | $\sim 9300$ | 811.03 | 21.68 | **30.31** | 46 |

[1] Independence test-based methods.
[2] Average running time evaluated on synthetic datasets containing 100 pairs, and each pair has 1000 samples.

## 5.2 BIVARIATE CAUSAL DISCOVERY

We evaluated the ROC-AUC score of bivariate causal discovery on both synthetic and real datasets.

**Datasets:** *Synthetic:* The generated datasets all follow the PNL model. Concretely, we considered the following two settings: 1) PNL-A: $f_1$ are general nonlinear functions generated by polynomials with random coefficients; and $f_2$ are monotonic nonlinear functions generated by unconstrained monotonic neural networks (UMNN) (Wehenkel & Louppe, 2019); 2) PNL-B: Both $f_1$ and $f_2$ are monotonic functions, generated by UMNN. The variances of $f_1, f_2$ are rescaled to 1. The input variable $X$ is sampled either from Gaussian mixture (mixG) or uniform (unif) distribution, and the injected noise $\epsilon$ is generated from normal distributions $N(0, \text{ns}^2)$, where $\text{ns} \in \{0.2, 0.4, 0.6, 0.8, 1\}$. Each configuration contains 100 data pairs, and each data pair has 1000 samples. *Gene Data:* Discovering gene-gene causal relationships is one important application. We pick the data used in DREAM4 competition (D4-S1,D4-S2A,D4-S2B,D4-S2C) (Marbach et al., 2009).

**Baselines & Evaluation:** Thanks to the implementation by Kalainathan et al. (2020), we can easily compare our proposed method with various existing algorithms. In this paper, we compared our proposed algorithm on both synthetic and real datasets with several baselines, including ANM (Hoyer et al., 2008), CDS (Fonollosa, 2019), IGCI (Janzing et al., 2012), RECI (Blöbaum et al., 2018), CDCI (Duong & Nguyen, 2022), OT-PNL (Tu et al., 2022), AbPNL (Uemura & Shimizu, 2020), and rank-PNL (Keropyan et al., 2023). Our implementation of MC-PNL follows Algorithm 1 (without fine-tuning), and we empirically set $\lambda = 5$ (the choice of $\lambda$ is briefly discussed in suppl. I.4). We compared two dependence measures RDC and HSIC-RBF for the ablation study. We also conducted causal discovery based on the PNL functions learned by the ACE algorithm for comparison. The causal scores $C_{X \to Y}$ calculated for each data pair are used for the ROC-AUC evaluation.

We report the comparison of ROC-AUCs in Table 2. The results are averaged over five different noise scales for the synthetic datasets. Our proposed MC-PNL is competitive compared to other independence test-based methods on the synthetic PNL data. Especially compared with AbPNL, our MC-PNL is **not sensitive to the initializations** and is **much more efficient ($300\times$ faster)**; compared to ACE (without independence regularizer), MC-PNL has better causal discovery accuracy. The rank-PNL method achieves good performance when the input $X$ is uniformly distributed, but fails when the input distribution gets closer to reality. The IGCI method also performs well on synthetic datasets, however, it cannot provide transparent and interpretable transformations as MC-PNL does. For gene datasets, our method is quite competitive and shows its good potential. We also note increasing the sample size can also improve the performance of MC-PNL, see suppl. I.5.

## 6 CONCLUSIONS

In this paper, we focus on the PNL model learning and propose a maximal correlation-based method, which can recover the nonlinear transformations accurately and swiftly. The key is to introduce the maximal correlation to avoid learning random independent noise. The proposed MC-PNL is more reliable than previous methods solely based on the independence loss. Besides the PNL model learning, we conduct experiments on the downstream causal discovery task where MC-PNL is superior to the SOTA independence test-based methods.

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

## A    DISCUSSION ON MI MINIMIZATION

It was shown that minimizing the MI, i.e. $\min_{f_{(\to)},g_{(\to)}} \mathrm{MI}\left(r_{(\to)}; x\right)$, is equivalent to maximizing $\mathbb{E}\log p\left(r_{(\to)}\right) + \mathbb{E}\log\left|\frac{d}{dy}g_{(\to)}(y)\right|$ (Zhang & Hyvärinen, 2009), where $p(\cdot)$ is the noise density assumed to be known. We find this objective interpretable since the first term, $\mathbb{E}\log p\left(r_{(\to)}\right)$, can be understood as the data-fitting term. The second term, $\mathbb{E}\log\left|\frac{d}{dy}g_{(\to)}(y)\right|$, can be understood from an information-geometric perspective (Daniušis et al., 2010). However, the equivalent maximization formula requires a known noise distribution to calculate the log-likelihood. Some works (Ma et al., 2020; Uemura & Shimizu, 2020) have been proposed to avoid this difficulty by using the HSIC instead of the MI.

## B    CONNECTIONS BETWEEN VICREG AND SOFT-HGR

Variance-Invariance-Covariance Regularization (VICReg) (Bardes et al., 2022) has shown an outstanding performance in self-supervised learning. When the dimension of the representation vectors (i.e., $f$ and $g$) reduces to one, the covariance term is vanished, and the VICReg objective becomes,

$$\min_{f,g}\quad \underbrace{\mathbb{E}(f(X)-g(Y))^2}_{\text{invariance term}} + \lambda\underbrace{\left[\mathrm{ReLU}(\gamma - \mathrm{var}(f(X))) + \mathrm{ReLU}(\gamma - \mathrm{var}(g(Y)))\right]}_{\text{variance term}}, \quad (16)$$

where $\lambda, \gamma > 0$ are the hyper-parameters that need to be tuned. We notice that it shares similar rationale with the HGR maximal correlation. To better spot that, we rewrite the Soft-HGR in (8) as,

$$\min_{f,g}\quad \underbrace{\mathbb{E}\left[f(X)-g(Y)\right]^2}_{\text{invariance term}} + \underbrace{\mathrm{var}(f(X))\,\mathrm{var}(g(Y)) - \mathrm{var}(f(X)) - \mathrm{var}(g(Y))}_{\text{variance term}}, \quad (17)$$
$$\text{s.t.}\quad \mathbb{E}[f(X)] = \mathbb{E}[g(Y)] = 0,$$

For both of them, the invariance term encourages the alignment of the learned features. The variance term encourages a $\gamma$-bounded variation (VICReg) or a variance-product ($\mathrm{var}(f(X))\,\mathrm{var}(g(Y))$) controlled variation (Soft-HGR) to avoid trivial solutions like $f(X) = g(Y) = $ constant.

## C    EXPERIMENTS ON MINIMIZING HSIC SOLELY

In this section, we show the PNL model learning result by solving $\min_{f,g}\widehat{\mathrm{HSIC}}(x, r) = \frac{1}{n^2}\mathrm{tr}(K_{xx}HL_{rr}H)$. We generated two synthetic datasets from the PNL model, $Y = f_2\left(f_1(X) + \epsilon\right)$, and each contains 1000 data samples. The data generation mechanisms are as follows (see Figure 4),

- Syn-1: $f_1(X) = X^{-1} + 10X, f_2(Z) = Z^3, X \sim U(0.1, 1.1), \epsilon \sim U(0, 5),$
- Syn-2: $f_1(X) = \sin(7X), f_2(Z) = \exp(Z), X \sim U(0, 1), \epsilon \sim N(0, 0.3^2).$

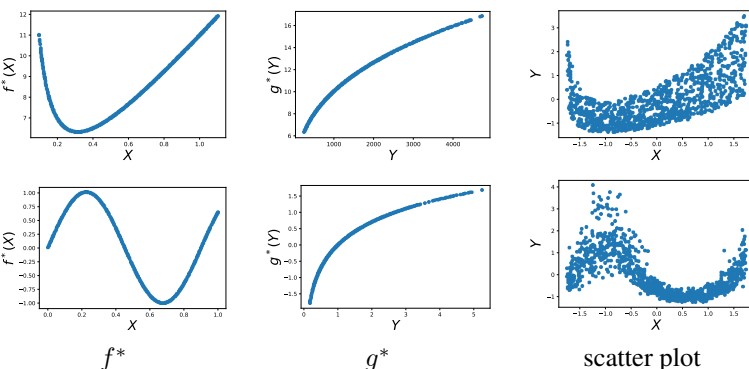

$$f^* \qquad\qquad g^* \qquad\qquad \text{scatter plot}$$

Figure 4: The ground truth transformations of $f^*$ and $g^*$ of Syn-1 (top) and Syn-2 (bottom).

We build different MLPs with the following configurations.

- **Narrow deep MLP**: Both the input and the output are one-dimensional; there are 9 hidden layers, each with 5 neurons. The activation function is `Leaky-ReLU`.
- **Wide over-parameterized MLP**: Both the input and the output are one-dimensional; there is only one single hidden layer with 9000 neurons. The activation function is `Leaky-ReLU`.

We use the default initialization method in PyTorch (Paszke et al., 2019), and make sure the initializations for all the narrow (or wide) MLPs are the same across experiments, as shown in Figure 5.

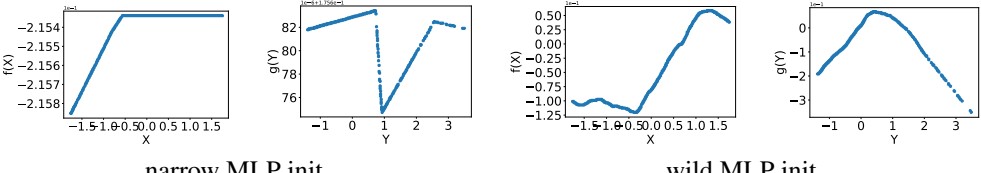

Figure 5: The initializations of MLPs.

**Optimization Setup:** We set the batch size to be 32. We use `Adam` (Kingma & Ba, 2015) for the optimization (the learning rates are $10^{-3}$ and $10^{-6}$ for narrow deep and wide over-parameterized MLPs, respectively, while all the other parameters are set by default).

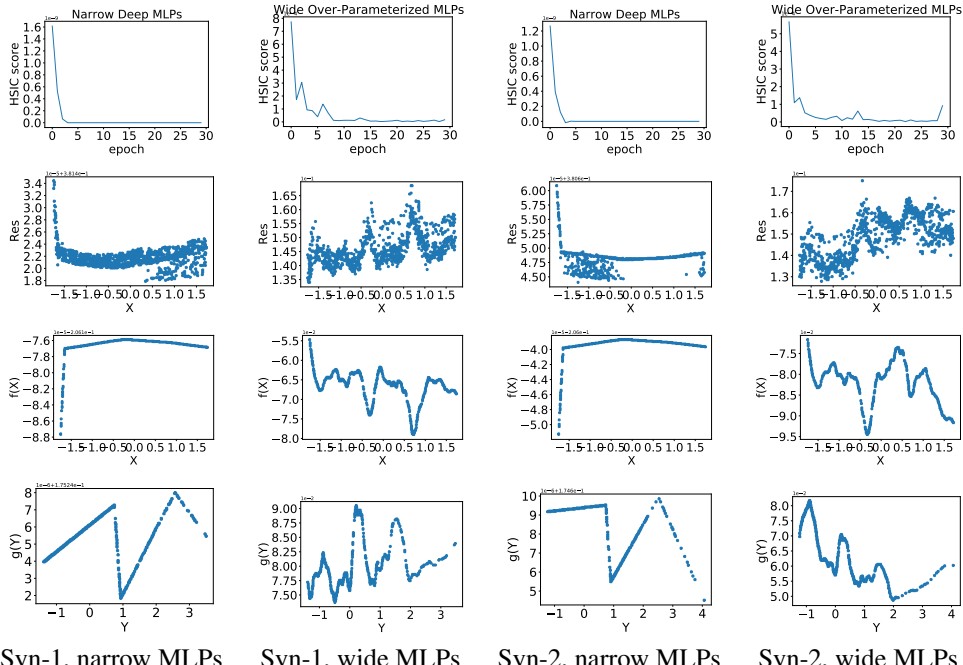

Figure 6: Visualization of the learned nonlinearities (trained solely with HSIC, under different dataset/MLP configurations). From top to bottom, the **convergence results**, **residual plot**, **learned** $f$, and **learned** $g$ are plotted, respectively. Each column shows one specific configuration of datasets and MLPs. **None of them learns meaningful nonlinearities**, and the learned transformations are quite similar across datasets.

We report the corresponding learning outcome in Figure 6. The learned transformations (see row 3 and row 4 in Figure 6) deviate far from the underlying ground truth functions, and are quite similar to each other. This is possibly because the solutions were trapped at the local minima near the same initialization point.

To verify whether such an HSIC-based PNL learning algorithm is stable for causal discovery, we further evaluate the baselines on the following dataset. We build 100 data pairs with different random seeds, following the same mechanism of Syn-1, and each contains 1000 data samples. We use

different MLP initializations for each data pair. The ROC-AUC scores reported in Table 3 show that the causal discovery stableness for ANM, IGCI, and AbPNL is not satisfactory.

Table 3: Comparison of bivaraite causal discovery ROC-AUC on 100 realizations of Syn-1.

| Dataset | ANM | CDS | IGCI | RECI | CDCI | AbPNL | ACE | MC-PNL |
|---------|-----|-----|------|------|------|-------|-----|--------|
| Syn-1 | 0.495 | 1 | 0.528 | 1 | 1 | 0.281 | 1 | 1 |

## D    DETAILED DATA DESCRIPTIONS

In this section, we describe the datasets in detail.

**Synthetic Datasets for Independence Test:**

In this section, we describe the synthetic data generation from PNL model for the independent test. The data were generated from the following model, $Y = f_2\left(f_1(X) + \epsilon\right), X \sim \text{GMM}, \epsilon \sim N(0, \sigma_\epsilon^2)$, where $f_1, f_2$ are randomly initialized monotonic neural networks (Wehenkel & Louppe, 2019) with 3 layers and 100 integration steps, and each layer contains 100 units. The cause term $X$ is sampled from a Gaussian mixture model as described in Lopez-Paz et al. (2017). The datasets were configured with various noise levels and sample sizes. There are three different injected noise levels, $\sigma_\epsilon \in \{0.01, 0.1, 1, 10\}$, and three different sample sizes, $N \in \{1000, 2000, 5000\}$. And under each configuration, we generated 100 data pairs for evaluating the independence test accuracy.

**Gene Datasets:**

For D4-S1, D4-S2A, D4-S2B, D4-S2C, we used the preprocessed data in Duong & Nguyen (2022) [1]. D4-S1 contains 36 variable pairs with 105 samples in each pair; D4-S2A, D4-S2B, D4-S2C contain 528, 747, and 579 variable pairs respectively, and each pair contains 210 samples.

## E    A UNIVERSAL VIEW OF DEPENDENCE MEASURES

Actually the discussed dependence measures in Section 3 are all closely related to the *mean squared contingency* introduced by (Rényi, 1959) and rediscovered due to its squared version called *squared-loss mutual information* (SMI) (Suzuki et al., 2009),

$$\text{SMI} := \iint p(x)p(y)\left(\frac{p(x,y)}{p(x)p(y)} - 1\right)^2 \mathrm{d}x\mathrm{d}y = \iint \frac{p(x,y)}{p(x)p(y)}p(x,y)\mathrm{d}x\mathrm{d}y - 1. \quad (18)$$

When the density ratio $\text{DR}(x,y) := \frac{p(x,y)}{p(x)p(y)}$ is a constant 1 (namely $X$ and $Y$ are independent), the SMI should be zero. To estimate the SMI, one can first approximate $\text{DR}(x,y)$ by a surrogate function $\text{DR}_{\boldsymbol{\theta}}(x,y)$ parameterized by $\boldsymbol{\theta}$, where the optimal parameter $\hat{\boldsymbol{\theta}}$ can be obtained via minimizing the following squared-error loss $J^{\text{DR}}$,

$$\begin{aligned} J^{\text{DR}}(\boldsymbol{\theta}) &:= \iint \left(\text{DR}_{\boldsymbol{\theta}}(x,y) - \text{DR}(x,y)\right)^2 p(x)p(y)\mathrm{d}x\mathrm{d}y \\ &= \iint \text{DR}_{\boldsymbol{\theta}}(x,y)^2 p(x)p(y)\mathrm{d}x\mathrm{d}y - 2\iint \text{DR}_{\boldsymbol{\theta}}(x,y)p(x,y)\mathrm{d}x\mathrm{d}y + \text{Const.} \end{aligned} \quad (19)$$

Then the empirical SMI can be calculated as, $\widehat{\text{SMI}} = \frac{1}{n}\sum_{j=1}^{n} \text{DR}_{\hat{\boldsymbol{\theta}}}(x_j, y_j) - 1$.

We show that, with different parameterizations of the density ratio, the resulting SMI will be equivalent to different dependence measures, see Table 4.

---

[1] https://github.com/baosws/CDCI

Table 4: Connections between DR parameterization and dependence measure

| Density ratio surrogate function $\mathrm{DR}_{\boldsymbol{\theta}}(x, y)$ | Corresponding dependence measure |
|---|---|
| $\mathrm{DR}_{\boldsymbol{\theta}}(x, y) = 1 + \sum_{i=1}^{n} \theta_i k\left(x, x_i\right) l\left(y, y_i\right)$ | variant of LSMI (Sugiyama & Yamada, 2012) |
| $\mathrm{DR}_{\boldsymbol{\theta}}(x, y) = 1 + \sum_{i=1}^{n} \frac{1}{n} k\left(x, x_i\right) l\left(y, y_i\right)$ | HSIC (Gretton et al., 2005) |
| $\mathrm{DR}_{\boldsymbol{\theta}}(x, y) = 1 + \sum_{i=1}^{m} f_i(x) g_i(y)$ | $m$-mode HGR correlation (Wang et al., 2019) |
| $\mathrm{DR}_{\boldsymbol{\theta}}(x, y) = 1 + f(x) g(y)$ [1] | HGR correlation (Rényi, 1959) |

[1] When $f, g$ are the linear combinations of random features, $f(x) = \boldsymbol{\alpha}^T \boldsymbol{\phi}(x), g(y) = \boldsymbol{\beta}^T \boldsymbol{\psi}(y)$, the corresponding dependence measure will be RDC (López-Paz et al., 2013),

In Sugiyama & Yamada (2012), they proposed to approximate the density ratio by $\mathrm{DR}_{\hat{\boldsymbol{\theta}}}(x, y) = \sum_{i=1}^{n} \hat{\theta}_i k\left(x, x_i\right) l\left(y, y_i\right)$, where $\hat{\boldsymbol{\theta}}$ has a closed-form solution via minimizing (19). After that, they approximated the SMI using the empirical average of Equation (18), $\frac{1}{n} \sum_{j=1}^{n} \mathrm{DR}_{\hat{\boldsymbol{\theta}}}(x_j, y_j) - 1 = \frac{1}{n} \sum_{j=1}^{n} \sum_{i=1}^{n} \hat{\theta}_i k\left(x, x_i\right) l\left(y, y_i\right) - 1$. It is shown that, the first term is actually the empirical HSIC, when $\{\hat{\theta}_i\}_{i=1}^{n} = \frac{1}{n}$.

*We argue that there is a small flaw in their derivation.* When $X$ and $Y$ are independent, both the SMI and the HSIC score should be zero. A simple modification is to model the density ratio by $\mathrm{DR}_{\boldsymbol{\theta}}(x, y) = 1 + \sum_{i=1}^{n} \theta_i k\left(x, x_i\right) l\left(y, y_i\right)$, see Table 4, where the constant 1 is to exclude all the independence terms, and the rest terms should model the dependency only. This modification will not hurt the quadratic form of $J^{\mathrm{DR}}(\boldsymbol{\theta})$, and maintains good interpretation. And the SMI is reduced to HSIC score when $\{\theta_i\}_{i=1}^{n} = \frac{1}{n}$.

We extend this density ratio estimation to a different parameterization, $\mathrm{DR}_{\boldsymbol{\theta}}(x, y) = 1 + f(x) g(y)$, where $f, g$ are zero mean and unit variance functions parameterized by $\boldsymbol{\theta}$. The resulting SMI will be equal to the HGR maximal correlation, see Proposition 2. Similarly, the constant 1 will capture the independence part, and $f(x) g(y)$ will capture the dependencies.

**Proposition 2.** *The density ratio estimation problem (19) is equivalent to the maximal HGR correlation problem, when the density ratio is modeled in the form of $\mathrm{DR}_{\boldsymbol{\theta}}(x, y) = 1 + f(x) g(y)$, and $f, g$ are restricted to zero mean and unit variance functions.*

*Proof.* We substitute $\mathrm{DR}_{\hat{\boldsymbol{\theta}}}(x, y)$ into Equation (19),

$$J^{\mathrm{DR}}(f, g) = \iint (1 + f(x) g(y))^2 p(x) p(y) \mathrm{d}x \mathrm{d}y - 2 \iint (1 + f(x) g(y)) p(x, y) \mathrm{d}x \mathrm{d}y + \mathrm{Const.}$$
$$= 1 + 2\mathbb{E}(f(X))\mathbb{E}(g(Y)) + \mathrm{var}(f(X))\mathrm{var}(g(Y)) - 2 - 2\mathbb{E}(f(X)g(Y)) + \mathrm{Const.}$$

Then it is not hard to see, $\min_{f,g} J^{\mathrm{DR}}(f, g)$, subject to $\mathbb{E}(f) = \mathbb{E}(g) = 0, \mathrm{var}(f) = \mathrm{var}(g) = 1$, is equivalent to the maximal HGR correlation problem. $\square$

**Proposition 3.** *The density ratio estimation problem (19) is equivalent to the Soft-HGR problem, when the density ratio is modeled in the form of $\mathrm{DR}_{\boldsymbol{\theta}}(x, y) = 1 + f(x) g(y)$, and $f, g$ are restricted to zero mean functions.*

We further note that the above density ratio estimation can be regarded as a truncated singular value decomposition $\mathrm{DR}_{\hat{\boldsymbol{\theta}}}(x, y) = 1 + \sum_{i=1}^{m} \sigma_i f_i(x) g_i(y)$, where $m = 1$ and $\sigma_i$ can be absorbed as scaling of functions (Buja, 1990). When letting $m > 1$ and imposing zero mean and orthonormal constraints on all $f_i$ and $g_i$, the corresponding $J^{\mathrm{DR}}$ minimization problem is equivalent to solving the $m$-mode HGR maximal correlation (Wang et al., 2019; Lee, 2021), as defined below.

**Definition 3** ($m$-mode HGR maximal correlation). *Given $1 \leq m \leq \min\{|\mathcal{X}|, |\mathcal{Y}|\}$, the $m$-mode maximal correlation problem for random variables $X \in \mathcal{X}, Y \in \mathcal{Y}$ is,*

$$(\mathbf{f}^*, \mathbf{g}^*) \triangleq \underset{\substack{\mathbf{f}:\mathcal{X} \rightarrow \mathbb{R}^m, \mathbf{g}:\mathcal{Y} \rightarrow \mathbb{R}^m \\ \mathbb{E}[\mathbf{f}(X)]=\mathbb{E}[\mathbf{g}(Y)]=\mathbf{0}, \\ \mathbb{E}[\mathbf{f}(X)\mathbf{f}^{\mathrm{T}}(X)]=\mathbb{E}[\mathbf{g}(Y)\mathbf{g}^{\mathrm{T}}(Y)]=\mathbf{I}}}{\arg\max} \mathbb{E}\left[\mathbf{f}^{\mathrm{T}}(X)\mathbf{g}(Y)\right], \tag{20}$$

where $\mathbf{f} = [f_1, f_2, \ldots, f_m]^{\mathrm{T}}, \mathbf{g} = [g_1, g_2, \ldots, g_m]^{\mathrm{T}}$ *are referred as the maximal correlation functions.*

## F THEORETICAL JUSTIFICATIONS OF THE PROPOSED METHODS

In this section, we prove the lemmas and propositions in our main paper in detail.

**Lemma 3** (Inconsistency of ACE). *Suppose the data were generated from the PNL model* $Y = f_2(f_1(X) + \epsilon)$, *where* $\epsilon \perp\!\!\!\perp X$. *Without loss of generality, we can get the ground truth transformation by rescaling,* $\bar{g}(Y) = \frac{f_2^{-1}(Y) - \mathbb{E}[f_2^{-1}(Y)]}{\sqrt{\mathbb{E}[f_2^{-1}(Y) - \mathbb{E}[f_2^{-1}(Y)]]^2}}$, *such that* $\mathbb{E}[g^*(Y)] = 0$ *and* $\mathbb{E}[g^*(Y)]^2 = 1$; *correspondingly,* $\bar{f}(X) = \frac{f_1(X) - \mathbb{E}[f_1(X)]}{\sqrt{\mathbb{E}[f_2^{-1}(Y) - \mathbb{E}[f_2^{-1}(Y)]]^2}}$ *with zero mean, and* $\bar{\epsilon} = \frac{\epsilon}{\sqrt{\mathbb{E}[f_2^{-1}(Y) - \mathbb{E}[f_2^{-1}(Y)]]^2}}$. *Then,* $(\bar{f}, \bar{g})$ *is not a local minimum of the ACE regression problem below,*

$$\min_{f,g} \mathbb{E}(f(X) - g(Y))^2, \quad s.t. \quad \mathbb{E}[f(X)] = \mathbb{E}[g(Y)] = 0, \quad \mathbb{E}[g^2(Y)] = 1. \tag{21}$$

*Proof.* If we fix $g = \bar{g}$ and optimize $f$, the optimization problem becomes

$$\min_{\mathbb{E}[f(X)]=0} \mathbb{E}(f(X) - \bar{g}(Y))^2 = \min_{\mathbb{E}[f(X)]=0} \mathbb{E}(f(X) - \bar{f}(X) + \bar{f}(X) - \bar{g}(Y))^2,$$
$$= \min_{\mathbb{E}[f(X)]=0} \mathbb{E}_{\bar{\epsilon},X}(f(X) - \bar{f}(X) - \bar{\epsilon})^2, \tag{22}$$

whose optimal solution align with the ground truth $\bar{f}$ (note that $\bar{\epsilon}$ is independent with $X$). While if we fix $f = \bar{f}$ and optimize $g$, the optimization problem becomes

$$\min_{\substack{\mathbb{E}[g(Y)]=0 \\ \mathbb{E}[g^2(Y)]=1}} \mathbb{E}(\bar{f}(X) - g(Y))^2 = \min_{\substack{\mathbb{E}[g(Y)]=0 \\ \mathbb{E}[g^2(Y)]=1}} \mathbb{E}(\bar{f}(X) - \bar{g}(Y) + \bar{g}(Y) - g(Y))^2,$$
$$= \min_{\substack{\mathbb{E}[g(Y)]=0 \\ \mathbb{E}[g^2(Y)]=1}} \mathbb{E}_{\bar{\epsilon},Y}(-\bar{\epsilon} + \bar{g}(Y) - g(Y))^2. \tag{23}$$

In this case, we **do not have** the independence between $\bar{\epsilon}$ and $Y$, so the optimal solution of (23)**would not be** $\bar{g}$. $\square$

The independence between the residual and input is not ensured for (21), so we can observe the distortion while applying the ACE algorithm solely. The independence constraint $f(X) - g(Y) \perp\!\!\!\perp X$ should be included to fix the above inconsistency, e.g., $\mathrm{HSIC}(f(X) - g(Y), X) = 0$, see (24).

$$\begin{aligned} \min_{f,g} \quad & \mathbb{E}(f(X) - g(Y))^2, \\ \text{s.t.} \quad & \mathbb{E}[f(X)] = \mathbb{E}[g(Y)] = 0, \mathbb{E}[g^2(Y)] = 1, \\ & \mathrm{HSIC}(f(X) - g(Y), X) = 0 \end{aligned} \tag{24}$$

Let us first show the model identifiability with the independence constraint in Lemma 4

**Lemma 4.** *Assuming invertible* $\bar{f}$ *and* $\bar{g}$, *we have*

$$\mathrm{HSIC}(f(X) - g(Y), X) = 0 \quad \Rightarrow \quad f(X) = a\bar{f}(X) + b_1 \text{ and } g(Y) = a\bar{g}(Y) + b_2, \tag{25}$$

*where* $a, b_0, b_1$ *are some constant numbers.*

To prove this, we can directly apply the identifiability results of post-nonlinear mixtures (Achard & Jutten, 2005) in the nonlinear independent component analysis (ICA) literature. We first reformulate the PNL learning problem as a two-dimensional blind source PNL mixture separation problem as in (26). Starting from the right-hand side, the independent sources are linearly mixed by a matrix $\boldsymbol{A}$, and then processed with component-wise invertible PNL functions (e.g., $\bar{f}^{-1}, \bar{g}^{-1}$ in (26)). One can observe the PNL mixed signals $(X, Y)^T$ in the middle. Taleb & Jutten (1999) proposed to separate out the independent sources through a separation structure consisting of un-mixing component-wise

nonlinearities (e.g., $\hat{f}, \hat{g}$ in (26)) and a separation matrix $\boldsymbol{B}$. Note that the matrices $\boldsymbol{A}, \boldsymbol{B}$ are usually unknown for the PNL mixture blind separation, but are given in our bivariate PNL learning context.

$$\begin{pmatrix} \hat{f}(X) \\ \hat{\epsilon} \end{pmatrix} \leftarrow \underbrace{\begin{pmatrix} 1 & 0 \\ 1 & -1 \end{pmatrix}}_{=:\boldsymbol{B}} \underbrace{\begin{pmatrix} \hat{f}(X) \\ \hat{g}(Y) \end{pmatrix} \xleftarrow[\hat{g}]{\hat{f}}}_{\text{separation structure}} \begin{pmatrix} X \\ Y \end{pmatrix} \xleftarrow[\bar{g}^{-1}]{\bar{f}^{-1}} \underbrace{\begin{pmatrix} 1 & 0 \\ 1 & 1 \end{pmatrix}}_{=:\boldsymbol{A}} \underbrace{\begin{pmatrix} \bar{f}(X) \\ \bar{\epsilon} \end{pmatrix}}_{\text{PNL mixture}} \tag{26}$$

Achard & Jutten (2005) showed that, under some weak assumptions, the separation mechanism in (26) can produce mutually independent components (e.g., independent $\hat{f}(X), \hat{\epsilon}$), **if and only if** the composition of PNL functions and un-mixing nonlinearities, $h_i$ (e.g., $h_1 = \hat{f} \circ \bar{f}^{-1}$, $h_2 = \hat{g} \circ \bar{g}^{-1}$), are linear, and $\boldsymbol{BA} = \boldsymbol{PD}$, where $\boldsymbol{P}$ is a permutation matrix, and $\boldsymbol{D}$ is a diagonal matrix. Now, we are prepared to see the proof of Lemma 4.

*Proof.* To make $\mathrm{HSIC}(f(X) - g(Y), X) = 0$, i.e., independent $\hat{\epsilon}$ and $\hat{f}(X)$, we need linear $h_i$ and $\boldsymbol{BA} = \boldsymbol{PD}$. The second condition $\boldsymbol{BA} = \boldsymbol{PD}$ is easy to check, since $\boldsymbol{BA} = \begin{pmatrix} 1 & 0 \\ 0 & -1 \end{pmatrix}$. To ensure linear $h_i$, we have

$$h_1(z) = \hat{f}(\bar{f}^{-1}(z)) = a_1 z + b_1 \xrightarrow{t=\bar{f}^{-1}(z)} \hat{f}(t) = a_1 \bar{f}(t) + b_1, \tag{27}$$

$$h_2(z) = \hat{g}(\bar{g}^{-1}(z)) = a_2 z + b_2 \xrightarrow{t=\bar{g}^{-1}(z)} \hat{g}(t) = a_2 \bar{g}(t) + b_2. \tag{28}$$

Only when $a_1 = a_2 = a$, the independence between $\hat{f}(X)$ and $\hat{f}(X) - \hat{g}(Y)$ is achieved. $\qquad\square$

**Corollary 1.** *Assuming invertible $f_1$ and $f_2$, the ground truth transformations $\pm(\bar{f}, \bar{g})$ are the **only two feasible solutions** of (24).*

*Proof.* According to Lemma 4, it is not hard to see the following results:

- $\mathbb{E}(g(Y)) = \mathbb{E}(a_1 \bar{g}(Y) + b_2) = 0 \quad \Rightarrow \quad b_2 = 0$,

- similarly, $\mathbb{E}(f(X)) = \mathbb{E}(a_1 \bar{f}(X) + b_1) = 0 \quad \Rightarrow \quad b_1 = 0$,

- $\mathbb{E}(g^2(Y)) = \mathbb{E}(a_1 \bar{g}(Y))^2 = 1 \quad \Rightarrow \quad a_1 = \pm 1$.

So there are only two feasible solutions. $\qquad\square$

A similar result holds for the Soft-HGR objective with independence constraint,

$$\begin{aligned} \min_{f,g} \quad & -\mathbb{E}\left[f(X)g(Y)\right] + \tfrac{1}{2} \operatorname{var}(f(X)) \operatorname{var}(g(Y)), \\ \text{s.t.} \quad & \mathbb{E}[f(X)] = \mathbb{E}[g(Y)] = 0, \\ & \mathrm{HSIC}(X, f(X) - g(Y)) = 0. \end{aligned} \tag{29}$$

**Corollary 2** (Proposition 1 in main paper). *Assuming invertible $f_1$ and $f_2$, the ground truth transformations $\pm(a^*\bar{f}, a^*\bar{g})$ are the **only two solutions** of (29).*

*Proof.* According to Lemma 4, the zero mean constraints $\mathbb{E}[f(X)] = \mathbb{E}[g(Y)] = 0$ imply $b_1 = b_2 = 0$. And we obtain optimal $a^* = \arg\min_a -a^2 \mathbb{E}\left[\bar{f}(X)\bar{g}(Y)\right] + \frac{1}{2} a^4 \mathbb{E}(\bar{f}^2(X))\mathbb{E}(\bar{g}^2(Y)) = \pm\sqrt{\frac{\mathbb{E}\left[\bar{f}(X)\bar{g}(Y)\right]}{\mathbb{E}(\bar{f}^2(X))\mathbb{E}(\bar{g}^2(Y))}}$. $\qquad\square$

The main purpose of introducing the Soft-HGR objective is to learn meaningful transformations rather than those in Figure 6. Also, the reformulation of (29) to its soft version (i.e., treat the hard constraint as a penalty term) allows efficient BCD-like optimization algorithms to be exploited.

## G   RANDOM FOURIER FEATURE GENERATION

We design a $k$-dimensional random feature vector $\boldsymbol{\phi}(x) = [\sin(w_1 x + b_1), \cdots, \sin(w_k x + b_k)]^T$, where $w_i, b_i \sim N(0, s^2)$. The random feature matrix $\Phi \in \mathbb{R}^{k \times n}$ is stacked as,

$$\Phi(\boldsymbol{x}; k, s) := \begin{pmatrix} \sin(w_1 x_1 + b_1) & \cdots & \sin(w_1 x_n + b_1) \\ \vdots & \vdots & \vdots \\ \sin(w_k x_1 + b_k) & \cdots & \sin(w_k x_n + b_k) \end{pmatrix}.$$

The same procedure can be applied to $\boldsymbol{y}$ as well to generate $\Psi$. The number of random Fourier features $k$ is user-defined, which is typically chosen from a few tens to a few thousands (Rahimi & Recht, 2008; Theodoridis, 2015). In our experiments, we set $k = 30$ and $s = 2$.

## H   ON THE OPTIMIZATION OF PROPOSED OBJECTIVE

### H.1   SUBPROBLEM: EQUALITY CONSTRAINED QUADRATIC PROGRAMMING

To simplify the notation, we rewrite the subproblem into the following form,

$$\begin{aligned} \min_{\boldsymbol{x} \in \mathbb{R}^n} \quad & f(\boldsymbol{x}) := \tfrac{1}{2} \boldsymbol{x}^T \boldsymbol{P} \boldsymbol{x} - \boldsymbol{q}^T \boldsymbol{x}, \\ s.t. \quad & \boldsymbol{v}^T \boldsymbol{x} = c. \end{aligned} \tag{30}$$

With the KKT conditions, one can find the unique optimal solution $\boldsymbol{x}^*$ by solving the following linear system,

$$\underbrace{\begin{pmatrix} \boldsymbol{P} & \boldsymbol{v} \\ \boldsymbol{v}^T & 0 \end{pmatrix}}_{=:\text{KKT}} \begin{pmatrix} \boldsymbol{x}^* \\ \lambda^* \end{pmatrix} = \begin{pmatrix} \boldsymbol{q} \\ c \end{pmatrix}, \tag{31}$$

when the KKT matrix is non-singular. In our setting, we can choose $\Phi$ and $\Psi$ properly to make $\Phi\Phi^T$ and $\Psi\Psi^T$ positive definite, or add a small positive definite perturbation matrix $\epsilon \boldsymbol{I}$, such that the unique optimum would be obtained. Besides, the sub-problem is of smaller size and easy to solve.

### H.2   LANDSCAPE STUDY WITH HESSIAN

To simplify the notation, we rewrite

$$J(\boldsymbol{\alpha}, \boldsymbol{\beta}; A, B, C, D, E) = \boldsymbol{\alpha}^T A \boldsymbol{\alpha} \boldsymbol{\beta}^T B \boldsymbol{\beta} - \boldsymbol{\alpha}^T C \boldsymbol{\beta} + \boldsymbol{\alpha}^T D \boldsymbol{\alpha} + \boldsymbol{\beta}^T E \boldsymbol{\beta}, \tag{32}$$

where $A = \frac{1}{2n^2} \Phi\Phi^T$, $B = \Psi\Psi^T$, $C = \frac{1}{n}\Phi\Psi^T + \frac{\lambda}{(n-1)^2}\Phi H K_{\boldsymbol{xx}} H \Psi^T$, $D = \frac{\lambda}{(n-1)^2}\Phi H K_{\boldsymbol{xx}} H \Phi^T$, $E = \frac{\lambda}{(n-1)^2}\Psi H K_{\boldsymbol{xx}} H \Psi^T$. The corresponding Hessian is

$$\nabla^2 J(\boldsymbol{\alpha}, \boldsymbol{\beta}) = \begin{pmatrix} 2A\boldsymbol{\beta}^T B \boldsymbol{\beta} + 2D & A\boldsymbol{\alpha}\boldsymbol{\beta}^T B - C \\ B^T \boldsymbol{\beta} \boldsymbol{\alpha}^T A - C^T & 2B\boldsymbol{\alpha}^T A \boldsymbol{\alpha} + 2E \end{pmatrix}. \tag{33}$$

Now we can verify the property of the critical points via checking their Hessians numerically. One obvious critical point is the all-zero vector $\boldsymbol{0}$. From our experiments, the Hessian at $\boldsymbol{0}$ is mostly indefinite, as long as the convex regularization term $\lambda$ is not too huge, which means $\boldsymbol{0}$ **is a saddle point that is easy to escape**. In practice, the algorithm rarely converges to $\boldsymbol{0}$.

## I   ADDITIONAL EXPERIMENTS

### I.1   FINE-TUNE WITH THE BANDED LOSS / HSIC WITH UNIVERSAL KERNEL

Recall the designed **banded residual loss** as follows. The data samples are separated into $b$ bins $\{\boldsymbol{x}^{(i)}, \boldsymbol{y}^{(i)}\}_{i=1}^b$ according to the ordering of $X$, and we expect the residuals in those bins $\text{Res}_i =$

$f(\boldsymbol{x}^{(i)}) - g(\boldsymbol{y}^{(i)})$ to have the same distribution, see Figure 7. To this end, we adopt the empirical maximum mean discrepancy (MMD) (Gretton et al., 2012) as a measure of distributional discrepancy. The **banded residual loss** is defined as $\mathrm{band}^{(\mathrm{MMD})} := \sum_{i=1}^{b} \widehat{\mathrm{MMD}}(\mathrm{Res}_i, \mathrm{Res}_{all})$, where $\mathrm{Res}_{all} = f(\boldsymbol{x}) - g(\boldsymbol{y})$. Then we append this $\mu$-penalized banded loss,

$$\min_{\boldsymbol{\alpha},\boldsymbol{\beta}} \quad J(\boldsymbol{\alpha},\boldsymbol{\beta}) + \mu \cdot \mathrm{band}^{(\mathrm{MMD})}, \quad \text{s.t.} \quad \boldsymbol{\alpha}^T \Phi \mathbf{1} = \boldsymbol{\beta}^T \Psi \mathbf{1} = 0. \tag{34}$$

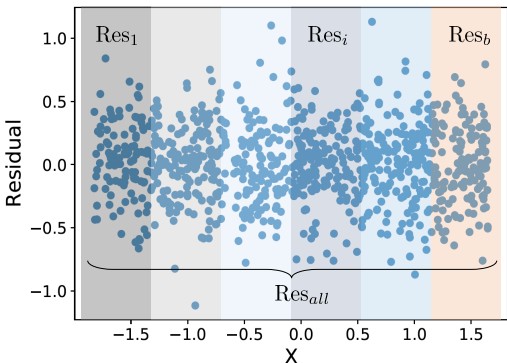

Figure 7: The construction of banded residual loss.

The above banded residual loss involves MMD, which is highly non-convex and brings difficulties to the optimization. We used the projected gradient descent with momentum to optimize (34). The residual plot shows a band shape, see the top row in Figure 8.

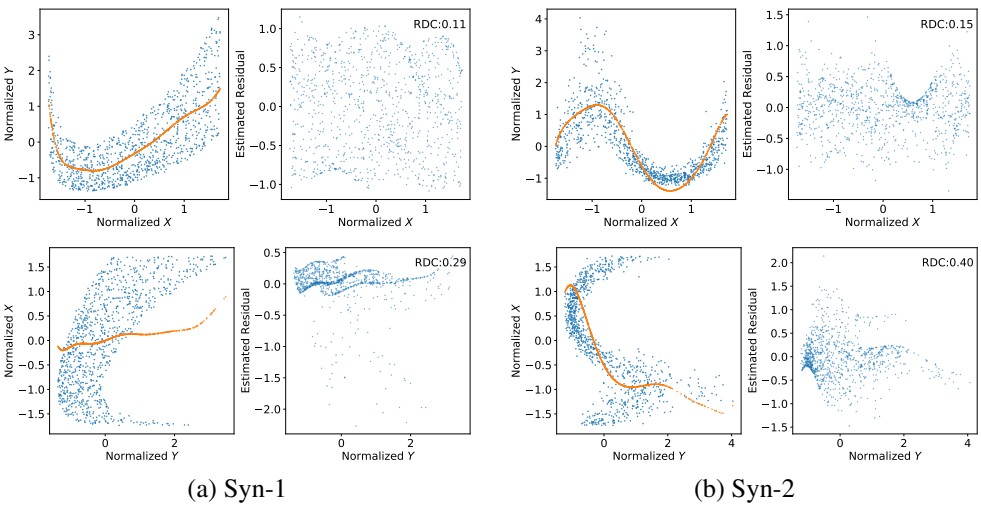

(a) Syn-1        (b) Syn-2

Figure 8: Fine-tuning with the banded residual loss.

We also show the results of fine-tuning by enlarging the penalty (to $\lambda = 10000$) HSIC term with the universal Gaussian RBF kernel in Figure 9.

**Definition 4** (Universal Kernel (Gretton et al., 2005)). *A continuous kernel $k(\cdot, \cdot)$ on a compact metric space $(\mathcal{X}, d)$ is called universal if and only if the RKHS $\mathcal{F}$ induced by the kernel is dense in $C(\mathcal{X})$, the space of continuous functions on $\mathcal{X}$, with respect to the infinity norm $\|f - g\|_\infty$.*

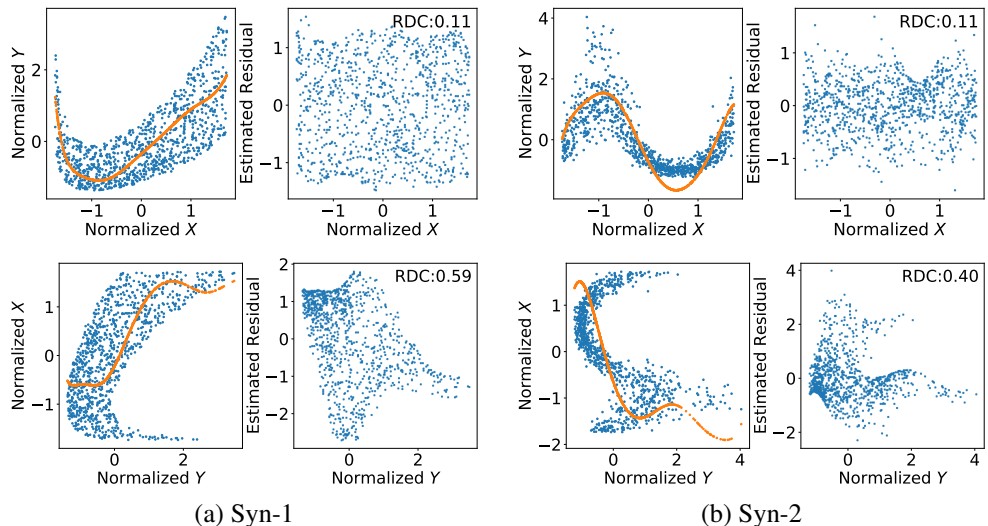

(a) Syn-1             (b) Syn-2

Figure 9: Fine-tuning with the HSIC-RBF loss.

## I.2 BOOTSTRAP FOR TRUSTWORTHY CAUSAL DISCOVERY

Bootstrap is a commonly used technique to estimate the confidence interval. In this section, we show a few examples of bootstrap with Tuebingen data (Mooij et al., 2016). We obtained 30 estimates of RDC from the data re-sampled with replacement, see Figure 10. The blue bars indicate the RDC distribution under the true causal direction; The orange bars indicate the RDC distribution under the false causal direction.

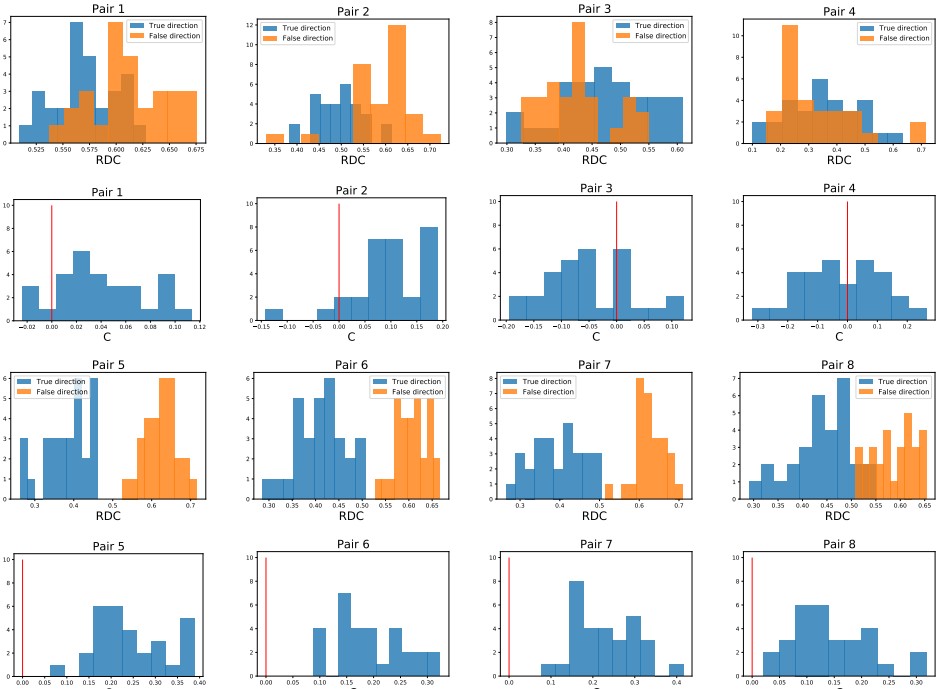

Figure 10: Bootstrap results of MC-PNL on eight Tuebingen data pairs. We plot the histogram of the RDC estimates and the estimated causal scores of 30 replications. The red vertical line indicates where $C_{X \to Y} = 0$.

### I.3 ADDITIONAL CONVERGENCE RESULTS

In this section, we show the convergence results on Syn-1 as well. With the regularization term (bottom in Figure 11), the algorithm can converge to two critical points with sign symmetry.

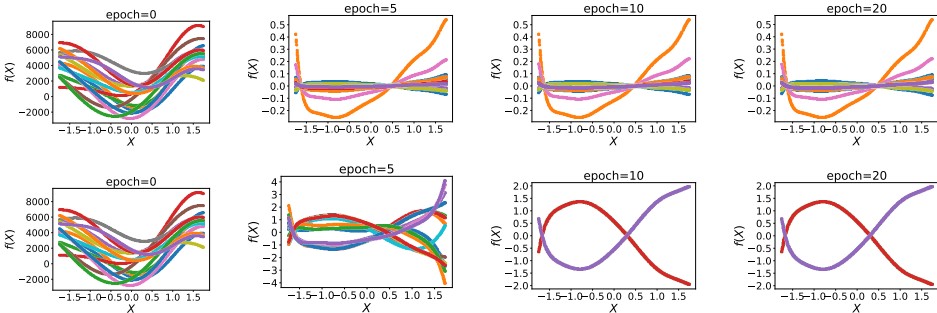

Figure 11: Convergence profile of Algorithm 1 on Syn-1.

### I.4 ON THE CHOICE OF $\lambda$

We tried seven different values for $\lambda$, and reported the AUC scores on the PNL-A-unif dataset with different noise levels. We found that the MC-PNL is suitable to use in the small noise regime. We also found that for the data with small noise, smaller $\lambda$ is preferred; and for the data with large injected noise, larger $\lambda$ is preferred.

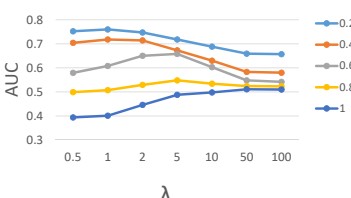

Figure 12: The detailed AUC scores vs. $\lambda$ under five noise levels on PNL-A-unif data.

### I.5 EFFECTS OF INCREASING THE SAMPLE SIZE

We show that the improvement of causal discovery accuracy when increasing the sample size in Table 5.

Table 5: The average AUC on synthetic datasets vs. sample size.

| sample size | 200 | 400 | 600 | 800 | 1000 | 5000 | 10000 |
|---|---|---|---|---|---|---|---|
| average AUC | 0.58935 | 0.6476 | 0.66915 | 0.6658 | 0.6738 | 0.7065 | 0.69785 |

