# OpenReview forum: "Post-Nonlinear Causal Relationship with Finite Samples: A Maximal Correlation Perspective"
_ICLR.cc/2024/Conference — Submitted to ICLR 2024_

### Official Review · Reviewer_hazx · 2023-10-29

**Soundness:** 3 good
**Presentation:** 3 good
**Contribution:** 2 fair
**Rating:** 6
**Confidence:** 3

**Summary:**

This paper studies the practical problem of the post-nonlinear model that focuses on the over-fitting issue and optimization issue in solving the non-linear function of PNL. The authors discuss several drawbacks of the independent test method, e.g., HSIC, and show that the randomized dependence coefficient (RDC) has the advantage in measuring the non-linear dependence, based on a set of simulation experiments. Moreover, The authors propose a novel method that incorporates maximal correlation into the PNL model learning (short as MC-PNL) such that the underlying nonlinearities can be accurately recovered. The experiment results verify the ability of non-linear function fitting of the proposed method and show that MC-PNL outperforms the baselines in causal discovery application.

**Strengths:**

1. The paper is well-written and easy to follow.

2. The authors gave a good overview on the related literature.

3. The authors provide a novel framework to deal with PNL learning, in which the maximal correlation constraints may be beneficial to fit the non-linear function of the PNL model.

**Weaknesses:**

1. There are extra assumptions to ensure the correctness of Lemma 4, such as "composition of PNL functions and un-mixing nonlinearities are linear" (provided in Proof Sec. F), which should be incorporated into the claim of Lemma 4.


2. The contribution of this paper is a proposed learning framework for PNL with some correctness analysis. My main concern is whether the theoretical contribution is insufficient due to the identification bound of PNL has not improved. I am willing to raise my score if the authors can further illustrate the signification impact of proposed methods in the fields of causal discovery.


3. In the "NONLINEAR FUNCTION FITTING" experiment, there are only two group generation mechanisms are used. More results with more types of non-linear functions should be provided if possible.

**Questions:**

Have you applied this method to real-world data?

---

> ### Author Response · Authors · 2023-11-15
>
> Thank you so much for your careful reading and comments. We have carefully addressed all your questions below and revised the manuscript accordingly. We sincerely hope that our answers can fully address your concerns.
>
> **W1: Presentation of Lemma 4. Should add more assumptions?**
>
> Lemma 4 is in the context of PNL learning and follows the same assumptions as Lemma 3. As we have assumed the **invertibility** of underlying functions $\bar{f}$ and $\bar{g}$ in Lemma 4, one can always find $\hat{f},\hat{g}$ such that " the composition of PNL functions and un-mixing nonlinearities are linear".
> We have added more descriptions to make Lemma 4 complete.
>
> **W2: Identification bound of PNL has not improved. Require illustration of the signification impact.**
>
> We acknowledge that the identification result has not been improved. However, we clarify the identifiability results in the context of PNL learning in Lemma 4, which has not been discussed in the existing literature. Besides, the estimation algorithm development is also fairly important, since there is a lack of effective and efficient algorithms to recover the underlying nonlinearities. Our proposed method is easy to implement and has a benign optimization structure, which provides a practical and efficient way to learn the underlying nonlinearities. We hope this can bring new thoughts to both the machine learning and causal inference communities.
>
> **W3: In the "NONLINEAR FUNCTION FITTING" experiment, there are only two group generation mechanisms are used. More results with more types of non-linear functions should be provided if possible.**
>
> The two synthetic examples were presented for illustration, and one of them was used in Uemura & Shimizu (2020). Following your suggestions, we updated the supplement with more examples. Please find additional experiments in our anonymous code link.
>
> **Q1: Have you applied this method to real-world data?**
> The gene data from the DREAM4 competition is designed to mimic real data. In addition, we also showed the causal discovery result with uncertainty quantification using a real dataset (Tuebingen) in Figure 10 in suppl.I.2.

---

> > ### Comment · Reviewer_hazx · 2023-11-16
> > **Thank you for the response.**
> >
> > Thank you for the response. Most of my concerns are addressed, so I will raise my score. I suggest the authors incorporate the responses to the final version of the paper if accepted.

---

> > > ### Author Response · Authors · 2023-11-17
> > >
> > > Thank you. We will certainly incorporate all the responses to the final version of our paper. We are always happy to hear any further questions or comments you may have.

---

### Official Review · Reviewer_7AyX · 2023-10-31

**Soundness:** 3 good
**Presentation:** 3 good
**Contribution:** 2 fair
**Rating:** 6
**Confidence:** 3

**Summary:**

This paper proposes a maximal correlation-based method for discovering causal relationships under the post-nonlinear causal model. Some theoretical guarantees are given. A few synthetic experiments are conducted as proof of concept. A simple real data analysis is also performed, comparing the proposed method against several existing benchmarks.

**Strengths:**

Overall, the paper is clearly written, with sufficient details on the motivation, theoretical results, and experiments.

The paper is trying to address an important question in causal inference, possibly drawing attention from fields such as causal inference, robust machine learning, and computational biology.

In general, causal discovery algorithms are difficult to scale up. However, this paper also discusses how to convexify the proposed algorithm so one can, at least in principle, solve the optimization problem to recover the final causal structure.

**Weaknesses:**

The numerical experiments look very simple (at least to me), and hence not entirely convincing about the use of overparameterized neural networks.

The optimization of over-parameterized neural nets itself, together with the "implicit regularization" effect of gradient-based methods, is a big problem in practice. I would hope that the authors discuss this in a remark.

When $f_{1}$ is a linear function in some basis of $X$, say $\beta^{\top} \phi (X)$, the proposed model is reduced to the single-index model. For single-index models, the identifiability of the parameter $\beta$ will be problematic, and often one assumes $\beta$ to be on the unit sphere. Is there a similar concern when one expands the modeling of $f_{1}$ to a purely nonlinear one?

I do not have much else to say about the "weaknesses" but I do want to mention that in the Questions section, I list several questions and comments on the paper.

**Questions:**

1. Why use maximal correlation? There are obviously many "nonparametric correlations" at our disposal, e.g. d-corr, maximal information coefficients (by Reshef et al.), and many others.

2. Is there any connection between the proposed method (or any other related method) and canonical correlation analysis (CCA)? From the formulation of the optimization problem alone, they look quite similar.

**Details Of Ethics Concerns:**

NA.

---

> ### Author Response · Authors · 2023-11-15
> **Response to Reviewer 7AyX**
>
> [ Update: Response to W2 is updated ]
>
> Thank you so much for your careful reading and comments. Our remarks and clarifications are as follows. We sincerely hope that our answers can fully address your concerns.
>
> **W1: The use of over-parameterized neural network (NN)**.
> We need to clarify that NN has been used in some previous PNL-based methods (e.g., MLP-PNL, AbPNL), but they are not used in our method. We observed the *over-fitting* and *optimization* issues when we tried to reproduce their results using NN, which motivates the benefits of using our proposed method based on the RFF. We did use different NN architectures (over-parameterized wide NN and a relatively smaller deep-narrow NN) to illustrate the issues of previous works, see supp. C, i.e., they both produce unsatisfying results.
>
> The reasons that we include the over-parameterized NN for comparison are as follows: 1) it works well in many practical applications, and 2) it has a nice loss landscape for optimization (no bad basin/spurious valley under mild assumptions [Sun et al. 2020]).
>
> **W2: In analogy with the single-index model, is there any identification issue when expanding $f_1$ to a purely nonlinear one?**
>
> Thanks for pointing out the relation to the single-index model. If we understood correctly, you are referring to the noiseless case, $y =m(x)= f_2(f_1(x))$, which is not discussed in our manuscript. When $f_1(x)= \beta^T\phi(x)$, the PNL model reduces to a single-index model as defined in [Lin and Kulasekera, 2007]. And $\beta$ is identifiable when it has a unit norm and $m$ is a non-constant continuous function.
>
>  If $f_1$ is extended to a more flexible nonlinear function w.r.t the parameter $\theta$ (e.g., a NN parameterized by $\theta$), similar results **MAY NOT** hold, i.e., $m(x) = g(f(x))=h(q(x))$ but $\{g\neq h, f\neq q \}$. A counter-example (in scalar case)  is $m(x)= \sin^2(x)$. The nonlinear transformations can be $\{g(z) = (\alpha z)^2,f(x)=\frac{1}{\alpha} \sin(x) \}$ and $\{h(z)= sin^2(\beta z), q(x) = \frac{1}{\beta} x \}$, where $\alpha,\beta \neq 0$ are some scaling factors.
>
>  However, in the PNL model in our paper, **there is an independent noise term**, and our Lemma 2 gives the corresponding identifiability result.
>
> Please correct us if we misunderstood your question. We'll be happy to address further questions/comments.
>
> **Q1: Why use maximal correlation? There are obviously many "nonparametric correlations" at our disposal, e.g. d-corr, maximal information coefficients (by Reshef et al.), and many others.**
>
> Maximal Correlation is a natural choice for PNL learning, as the optimized transformations $f$ and $g$ are closely related to the underlying nonlinear functions $f_1,f_2^{-1}$ in the PNL model. However, other dependence measures do not involve such a transformation learning process. Another consideration is computational efficiency. Since we need to compute dependence during optimization, many known dependence measures (e.g., maximal information coefficient) are not suitable, see the time comparison in the RDC paper (Lopez-Paz et al., 2013).  For the independence test, however, other dependence measures could be considered as potential replacements.
>
> **Q2: Any connection to CCA?**
> Certainly, some components in our method (e.g., HGR and RDC) are strongly related to nonlinear CCA. One famous work along this research line is Kernelized CCA by Bach and Jordan (2002), where the connection to CCA was thoroughly discussed. The distinction between Kernelized CCA and RDC lies in the representations of $f$ and $g$: Kernelized CCA uses $K_1\alpha_1$ for $f(X)$ (representer theorem), while RDC uses the linear combination of RFFs, $\Phi(X)^T\alpha$.
>
>
> ```
> R. Sun, D. Li, S. Liang, T. Ding and R. Srikant, "The Global Landscape of Neural Networks: An Overview," in IEEE Signal Processing Magazine, vol. 37, no. 5, pp. 95-108, Sept. 2020, doi: 10.1109/MSP.2020.3004124.
>
> Lin, Wei, and K. B. Kulasekera. “Identifiability of Single-Index Models and Additive-Index Models.” Biometrika 94, no. 2 (2007): 496–501. http://www.jstor.org/stable/20441387.
>
> Bach F R, Jordan M I. Kernel independent component analysis[J]. Journal of machine learning research, 2002, 3(Jul): 1-48.
>
> ```

---

> > ### Comment · Reviewer_7AyX · 2023-11-22
> > **thank you for your response**
> >
> > I thank the authors for their response to my comments. Unfortunately, I do not think I can improve my original evaluation of the paper, given the overall content of the paper, in terms of novelty and contribution.

---

### Official Review · Reviewer_m7U7 · 2023-11-01

**Soundness:** 2 fair
**Presentation:** 2 fair
**Contribution:** 2 fair
**Rating:** 6
**Confidence:** 3

**Summary:**

This paper considers bivariate causal discovery without confounders. Under the assumption that the true model follows the post-nonlinear (PNL) model, prior work does causal discovery by learning the functions f_1,f_2 and then using some dependence measure to compare the dependence between the residuals and the input under both hypotheses. This paper proposes an alternate two-stage method, where the first stage learns the functions using a soft-version of HGR maximal correlation regularized by a dependency measure (Renyi, 1959) and the second stage is an independence test between the residual and the input. The dependency measure regularization is motivated by pointing out the con of maximal correlation that it doesn't provide usable residuals for the downstream independence test. Experimental results show that the proposed method is competitive w.r.t existing methods.

**Strengths:**

Bivariate causal discovery is an important, fundamental problem in causal inference. This paper advances the literature by providing an algorithm that uses a variant of maximal correlation to learn nonlinear functions of the PNL model. A major contribution of this paper is a systematic study of how to use maximal correlation based methods for causal discovery. Experimental results seem comprehensive in the sense that they cover a wide variety of datasets both simulated and real, barring a few concerns that I elaborate in the following section.

**Weaknesses:**

1) The writing can be improved greatly. Few assertions are vague (e.g. "HSIC can get easily stuck at "meaningless" local minima", IGCI cannot provide "transparent and interpretable transformations") and undefined in the main paper (e.g. randomized dependence coefficient is not defined clearly despite it being used in the proposed method).
2) It was also not clear to me how a dependence measure, HGR correlation, was used to motivate learning nonlinear functions; after all it is irrelevant whether X and Y are dependent.
3) The main contributions seem overstated. Among a host of different causal disvoery methods compared in Table 2, the proposed method MC-PNL is faster compared to only AB-PNL by 300x, while there exists a competitive method in IGCI that is 60x faster than the proposed method. Overall, the experimental results don't seem to give the impression that MC-PNL outperforms other benchmarks.
4) While one of the disadvantages of the PNL algorithms is claimed to be the optimization issue, thus motivating the RFF parametrization and linear HSIC kernel, neither is the experimental performance of this variant discussed, nor is its theoretical properties. MC-PNL still uses a gradient-based algorithm for the universal kernel and banded loss regularizer.

**Questions:**

1) The authors repeatedly use the word "meaningful" (pg 5, under eq 9, pg 6, first para in 4.1) while criticizing existing methods without providing much explanation as to what is "meaningful". While the usage in pg 5 is backed by the observation that residuals can be matched to arbitrary noise profiles, other usages are unclear. These assertions seem important but without knowing the meaning it's unclear what the criticism means.
2) Is it supposed to be < -\delta in Line 3 in Algorithm 2?
3) Is there any ablation study done to determine the parameters of the RFF?

---

> ### Author Response · Authors · 2023-11-15
>
> Thank you so much for your careful reading and comments. We have carefully addressed your questions below and revised the manuscript accordingly. We sincerely hope that our answers have fully addressed your concerns.
>
> **W1 & Q1: On Presentation clarity.**
>
> We refer to the "meaningful solutions" as those solutions that can reflect the underlying nonlinear transformations. As one can see in the last two rows in Figure 6, the learned solutions cannot reveal the underlying functions as shown in Figure. 4, and thus are less meaningful. We have added a short description in the main paper to make this clear.
>
> The causal discovery method, IGCI, is not designed for PNL learning and cannot give explicit function transformations of $f_1,f_2$, thus cannot provide "transparent and interpretable transformations". We have rewritten this phrase to "explicit nonlinear transformations".
>
> For the convenience of the audience, we have added the formal definition of randomized dependence coefficient (RDC) in the updated version.
>
> **W2: How HGR was used to motivate PNL learning?**
> As $Y = f_2 (f_1(X)+\epsilon)$ in the PNL setting, it is obvious that $X$ and $Y$ are highly correlated when the noise is small. Our goal is to learn $f_1,f_2$. Let $f(X) = f_1(X),g(Y) = f_2^{-1}(Y)$, and we hope to learn $f$ and $g$ through maximizing the HGR correlation $E(g(Y)f(X))$.
>
> **W3: The main contributions seem overstated.**
> The reported numerical gain is compared with the SOTA PNL-based method. Our focus lies on comparing with interpretable baselines, as other baselines are incapable of providing explicit transformations for $f_1$ and $f_2$.
>
> **W4:  While one of the disadvantages of the PNL algorithms is claimed to be the optimization issue, thus motivating the RFF parametrization and linear HSIC kernel, neither is the experimental performance of this variant discussed, nor is its theoretical properties. MC-PNL still uses a gradient-based algorithm for the universal kernel and banded loss regularizer.**
>
> We did include the theoretical properties in the text under eq (14). We show that the BCD algorithm can converge to a critical point and each sub-problem is a quadratic program.
> The experimental convergence results have been shown in Figure 3. Only for fine-tuning, the gradient-based optimization is needed to handle the universal kernel and banded loss regularizer.
>
> **Q1: see W1**
>
> **Q2: Typo.**
> Yes, it should be `$<-\delta$` in Line 3 in Algorithm 2.
>
> **Q3: Is there any ablation study done to determine the parameters of the RFF?**
>
> We are unsure about the specific type of ablation study you are expecting. It would be very helpful if you could specify the question a little bit more.

---

> > ### Comment · Reviewer_m7U7 · 2023-11-20
> > **Reviewer Response to Authors**
> >
> > Thanks for the clarification about how HGR is used to motivate PNL learning and more clarity on the vague terms. For the former, it would make the paper clearer if your explanation were added.
> >
> > W3: I agree that it is competitive w.r.t other PNL-based methods. My concern is about the overstatement in the abstract and the introduction which doesn't mention about comparing with respect to PNL-methods there.
> >
> > Q3: Ablation: RFF has two parameters - noise variance and the number of sinusoidal features. I am referring to ablation studies w.r.t. these parameters on ROC-AUC.
> >
> > I have improved my score to reflect the responses.

---

> > > ### Author Response · Authors · 2023-11-21
> > >
> > > Thank you for the further comments and clarifying Q3 for us. Our responses are as follows.
> > >
> > > **W3:** We have adjusted the corresponding statements in the abstract and abstraction.
> > >
> > > **Q3: On the hyper-parameter selection of RFF (i.e., the number of RFFs $k$ and the variance factor $s$).** The parameters are empirically fixed as $k=30,s=2$ in our paper, and the corresponding model expressivity was checked on various continuous functions within a certain input range (note that $(X,Y)$ were standardized), see `syn_data_bivariate_causal_discovery
> > > genPNL_check_expressive.ipynb` in the anonymous code repo.
> > >
> > > From our experience, increasing the number of parameters $k$ slightly influences the ROC-AUC performance, however, brings a computational burden; tuning variance factor $s$ can bring a performance gain.  We report the AUC on one synthetic dataset in the following table for your reference.
> > >
> > > | k   | s   | AUC   | Time     |
> > > |-----|-----|-------|----------|
> > > | 30  | 0.5 | 0.628 | 43.56105 |
> > > | 60  | 0.5 | 0.64  | 80.27628 |
> > > | 120 | 0.5 | 0.656 | 234.0929 |
> > > | 30  | 2   | 0.692 | 46.71287 |
> > > | 60  | 2   | 0.685 | 80.79978 |
> > > | 120 | 2   | 0.691 | 236.0387 |
> > > | 30  | 4   | 0.73  | 45.18569 |
> > > | 60  | 4   | 0.725 | 80.4238  |
> > > | 120 | 4   | 0.712 | 237.8293 |

---

### Meta-Review · Area_Chair_a7aS · 2023-12-08

**Metareview:**

A new and interesting approach to bivariate causal discovery based on the post-nonlinear model is proposed. The authors use a variational formulation via the HGR maximal correlation to learn the nonlinearities, which converts the problem into an optimization problem that can be heuristically minimized by block coordinate descent. While these ideas are interesting, during review it was noted that the presentation was often vague and in some cases significantly overclaimed and overstated their results.

Reviewers flagged numerous examples of this. For one example, in the abstract, the authors write
> Owing to the benign structure of our objective function, when
modeling the nonlinearities with linear combinations of random Fourier features, the target optimization problem can be solved rather efficiently and rapidly via the block coordinate descent.

However, there is no proof of convergence and existing methods are in fact competitive with the proposed approach. In Proposition 1, the authors do show that there are two equivalent global minimizers, but there is no discussion of nonconvexity or local minima. In the appendix, the authors note that the method can get trapped at local minima. There are more examples in the reviews.

Since this was a borderline case, I took a careful look and evaluated these concerns myself. Unfortunately I find the issues to be serious, and in need of major revision and another round of review before the paper can be accepted.

**Justification For Why Not Higher Score:**

See meta-review

**Justification For Why Not Lower Score:**

N/A

---

### Decision · Program_Chairs · 2024-01-16

Reject